# Overinterpretation reveals image classification model pathologies

**Brandon Carter**
MIT CSAIL
bcarter@csail.mit.edu

**Siddhartha Jain**
MIT CSAIL

**Jonas Mueller**
Amazon Web Services

**David Gifford**
MIT CSAIL
gifford@mit.edu

## Abstract

Image classifiers are typically scored on their test set accuracy, but high accuracy can mask a subtle type of model failure. We find that high scoring convolutional neural networks (CNNs) on popular benchmarks exhibit troubling pathologies that allow them to display high accuracy even in the absence of semantically salient features. When a model provides a high-confidence decision without salient supporting input features, we say the classifier has overinterpreted its input, finding too much class-evidence in patterns that appear nonsensical to humans. Here, we demonstrate that neural networks trained on CIFAR-10 and ImageNet suffer from overinterpretation, and we find models on CIFAR-10 make confident predictions even when 95% of input images are masked and humans cannot discern salient features in the remaining pixel-subsets. We introduce Batched Gradient SIS, a new method for discovering sufficient input subsets for complex datasets, and use this method to show the sufficiency of border pixels in ImageNet for training and testing. Although these patterns portend potential model fragility in real-world deployment, they are in fact valid statistical patterns of the benchmark that alone suffice to attain high test accuracy. Unlike adversarial examples, overinterpretation relies upon unmodified image pixels. We find ensembling and input dropout can each help mitigate overinterpretation.

## 1 Introduction

Well-founded decisions by machine learning (ML) systems are critical for high-stakes applications such as autonomous vehicles and medical diagnosis. Pathologies in models and their respective training datasets can result in unintended behavior during deployment if the systems are confronted with novel situations. For example, a medical image classifier for cancer detection attained high accuracy in benchmark test data, but was found to base decisions upon presence of rulers in an image (present when dermatologists already suspected cancer) [1]. We define model *overinterpretation* to occur when a classifier finds strong class-evidence in regions of an image that contain no semantically salient features. Overinterpretation is related to overfitting, but overfitting can be diagnosed via reduced test accuracy. Overinterpretation can stem from true statistical signals in the underlying dataset distribution that happen to arise from particular properties of the data source (e.g., dermatologists' rulers). Thus, overinterpretation can be harder to diagnose as it admits decisions that are made by statistically valid criteria, and models that use such criteria can excel at benchmarks. We demonstrate overinterpretation occurs with unmodified subsets of the original images. In contrast to *adversarial examples* that modify images with extra information, overinterpretation is based on real patterns already present in the training data that also generalize to the test distribution. Hidden statistical signals of benchmark datasets can result in models that overinterpret or do not generalize to new data from a different distribution. Computer vision (CV) research relies on datasets like CIFAR-10 [2] and ImageNet [3] to provide standardized performance benchmarks. Here, we analyze the overinterpretation of popular CNN architectures on these benchmarks to characterize pathologies.

35th Conference on Neural Information Processing Systems (NeurIPS 2021).

Revealing overinterpretation requires a systematic way to identify which features are used by a model to reach its decision. Feature attribution is addressed by a large number of interpretability methods, although they propose differing explanations for the decisions of a model. One natural explanation for image classification lies in the set of pixels that is sufficient for the model to make a confident prediction, even in the absence of information about the rest of the image. In the example of the medical image classifier for cancer detection, one might identify the pathological behavior by finding pixels depicting the ruler alone suffice for the model to confidently output the same classifications. This idea of Sufficient Input Subsets (SIS) has been proposed to help humans interpret the decisions of black-box models [4]. An SIS subset is a minimal subset of features (e.g., pixels) that suffices to yield a class probability above a certain threshold with all other features masked.

We demonstrate that classifiers trained on CIFAR-10 and ImageNet can base their decisions on SIS subsets that contain few pixels and lack human understandable semantic content. Nevertheless, these SIS subsets contain statistical signals that generalize across the benchmark data distribution, and we are able to train classifiers on CIFAR-10 images missing 95% of their pixels and ImageNet images missing 90% of their pixels with minimal loss of test accuracy. Thus, these benchmarks contain inherent statistical shortcuts that classifiers optimized for accuracy can learn to exploit, instead of learning more complex *semantic* relationships between the image pixels and the assigned class label. While recent work suggests adversarially robust models base their predictions on more semantically meaningful features [5], we find these models suffer from overinterpretation as well. As we subsequently show, overinterpretation is not only a conceptual issue, but can actually harm overall classifier performance in practice. We find model ensembling and input dropout partially mitigate overinterpretation, increasing the semantic content of the resulting SIS subsets. However, this mitigation is not a substitute for better training data, and we find that overinterpretation is a statistical property of common benchmarks. Intriguingly, the number of pixels in the SIS rationale behind a particular classification is often indicative of whether the image is correctly classified.

It may seem unnatural to use an interpretability method that produces feature attributions that look uninterpretable. However, we do not want to bias extracted rationales towards human visual priors when analyzing a model's pathologies, but rather faithfully report the features used by a model. To our knowledge, this is the first analysis showing one can extract nonsensical features from CIFAR-10 and ImageNet that intuitively should be insufficient or irrelevant for a confident prediction, yet are alone sufficient to train classifiers with minimal loss of performance. Our contributions include:

- We discover the pathology of overinterpretation and find it is a common failure mode of ML models, which latch onto non-salient but statistically valid signals in datasets (Section 4.1).
- We introduce Batched Gradient SIS, a new masking algorithm to scale SIS to high-dimensional inputs and apply it to characterize overinterpretation on ImageNet (Section 3.2).
- We provide a pipeline for detecting overinterpretation by masking over 90% of each image, demonstrating minimal loss of test accuracy, and establish lack of saliency in these patterns through human accuracy evaluations (Sections 3.3, 4.2, 4.3).
- We show misclassifications often rely on smaller and more spurious feature subsets suggesting overinterpretation is a serious practical issue (Section 4.4).
- We identify two strategies for mitigating overinterpretation (Section 4.5). We demonstrate that overinterpretation is caused by spurious statistical signals in training data, and thus training data must be carefully curated to eliminate overinterpretation artifacts.

Code for this paper is available at: https://github.com/gifford-lab/overinterpretation.

## 2   Related Work

While existing work has demonstrated numerous distinct flaws in deep image classifiers our paper demonstrates a new distinct flaw, overinterpretation, previously undocumented in the literature. There has been substantial research on understanding dataset bias in CV [6, 7] and the fragility of image classifiers deployed outside benchmark settings. We extend previous work on sufficient input subsets (SIS) [4] with the Batched Gradient SIS method, and use this method to show that ImageNet sufficient input subset pixels for training and testing often exist at image borders. Many alternative interpretability methods also aim to understand models by extracting *rationales* (pixel-subsets) that

provide positive evidence for a class [8–11], and we adopt SIS throughout this work as a particularly straightforward method for producing such rationales. This prior work (including SIS [4]) is limited to understanding models and does not use the enhanced understanding of models to identify the overinterpretation flaw discovered in this paper. We contrast the issue of overinterpretation against other previously known model flaws below:

- Image classifiers have been shown to be fragile when objects from one image are transplanted in another image [12], and can be biased by object context [13, 14]. In contrast, overinterpretation differs because we demonstrate that highly sparse, unmodified subsets of pixels in images suffice for image classifiers to make the same predictions as on the full images.

- Lapuschkin et al. [15] demonstrate that DNNs can learn to rely on spurious signals in datasets, including source tags and artificial padding, but which are still human-interpretable. In contrast, the patterns we identify are minimal collections of pixels in images that are semantically meaningless to humans (they do not comprise human-interpretable parts of images). We demonstrate such patterns generalize to the test distribution suggesting they arise from degenerate signals in popular benchmarks, and thus models trained on these datasets may fail to generalize to real-world data.

- CNNs in particular have been conjectured to pick up on localized features like texture instead of more global features like object shape [16, 17]. Brendel and Bethge [18] show CNNs trained on natural ImageNet images may rely on local features and, unlike humans, are able to classify texturized images, suggesting ImageNet alone is insufficient to force DNNs to rely on more causal representations. Our work demonstrates another source of degeneracy of popular image datasets, where sparse, unmodified subsets of training images that are meaningless to humans can enable a model to generalize to test data. We provide one explanation for why ImageNet-trained models may struggle to generalize to out-of-distribution data.

- Geirhos et al. [19] discover that DNNs trained on distorted images fail to generalize as well as human observers when trained under image distortions. In contrast, overinterpretation reveals a different failure mode of DNNs, whereby models latch onto spurious but statistically valid sets of features in undistorted images. This phenomenon can limit the ability of a DNN to generalize to real-world data even when trained on natural images.

- Other work has shown deep image classifiers can make confident predictions on nonsensical patterns [20], and the susceptibility of DNNs to adversarial examples or synthetic images has been widely studied [5, 21–23]. However, these adversarial examples synthesize artificial images or modify real images with auxiliary information. In contrast, we demonstrate overinterpretation of unmodified subsets of actual training images, indicating the patterns are already present in the original dataset. We further demonstrate that such signals in training data actually generalize to the test distribution and that adversarially robust models also suffer from overinterpretation.

- Hooker et al. [24] found sparse pixel subsets suffice to attain high classification accuracy on popular image classification datasets, but evaluate interpretability methods rather than demonstrate spurious features or discover overinterpretation.

- Ghorbani et al. [25] introduce principles and methods for human-understandable concept-based explanations of ML models. In contrast, overinterpretation differs because the features we identify are semantically meaningless to humans, stem from single images, and are not aggregated into interpretable concepts. The existence of such subsets stemming from unmodified subsets of images suggests degeneracies in the underlying benchmark datasets and failures of modern CNN models to rely on more robust and interpretable signals in training datasets.

- Geirhos et al. [26] discuss the general problem of "shortcut learning" but do not recognize that 5% (CIFAR-10) or 10% (ImageNet) spurious pixel-subsets are statistically valid signals in these datasets, nor characterize pixels that provide sufficient support and lead to overinterpretation.

- In natural language processing (NLP), Feng et al. [27] explored model pathologies using a similar technique, but did not analyze whether the semantically spurious patterns relied on are a statistical property of the dataset. Other work has demonstrated the presence of various spurious statistical shortcuts in major NLP benchmarks, showing this problem is not unique to CV [28].

# 3 Methods

## 3.1 Datasets and Models

CIFAR-10 [2] and ImageNet [3] have become two of the most popular image classification benchmarks. Most image classifiers are evaluated by the CV community based on their accuracy in one of these benchmarks. We also use the CIFAR-10-C dataset [29] to evaluate the extent to which our CIFAR-10 models can generalize to out-of-distribution (OOD) data. CIFAR-10-C contains variants of CIFAR-10 test images altered by various corruptions (e.g., Gaussian noise, motion blur). Where computing sufficient input subsets on CIFAR-10-C images, we use a uniform random sample of 2000 images across the entire CIFAR-10-C set. Additional results on CIFAR-10.1 v6 [30] are presented in Table S4. We use the ILSVRC2012 ImageNet dataset [3].

For CIFAR-10, we explore three common CNN architectures: a deep residual network with depth 20 (ResNet20) [31], a v2 deep residual network with depth 18 (ResNet18) [32], and VGG16 [33]. We train these networks using cross-entropy loss optimized via SGD with Nesterov momentum [34] and employ standard data augmentation strategies [32] (Section S2). After training many CIFAR-10 networks individually, we construct four different ensemble classifiers by grouping various networks together. Each ensemble outputs the average prediction over its member networks (specifically, the arithmetic mean of their logits). For each of three architectures, we create a corresponding homogeneous ensemble by individually training five networks of that architecture. Each network has a different random initialization, which suffices to produce substantially different models despite having been trained on the same data [35]. Our fourth ensemble is heterogeneous, containing all 15 networks (5 replicates of each of 3 distinct CNN architectures).

For ImageNet, we use a pre-trained Inception v3 model [36] that achieves 22.55% and 6.44% top-1 and top-5 error [37] Additional results from an ImageNet ResNet50 are presented in Section S6.

## 3.2 Discovering Sufficient Features

**CIFAR-10.** We interpret the feature patterns learned by CIFAR-10 CNNs using the Sufficient Input Subsets (SIS) procedure [4], which produces rationales (SIS subsets) of a black-box model's decision-making. SIS subsets are minimal subsets of input features (pixels) whose values alone suffice for the model to make the same decision as on the original input. Let $f_c(x)$ denote the probability that an image $x$ belongs to class $c$. An SIS subset $S$ is a minimal subset of pixels of $x$ such that $f_c(x_S) \geq \tau$, where $\tau$ is a prespecified confidence threshold and $x_S$ is a modified input in which all information about values outside $S$ are masked. We mask pixels by replacement with the mean value over all images (equal to zero when images have been normalized), which is presumably least informative to a trained classifier [4]. SIS subsets are found via a local backward selection algorithm applied to the function giving the confidence of the predicted (most likely) class.

**ImageNet.** We scale the SIS backward selection procedure to ImageNet with the introduction of Batched Gradient SIS, a gradient-based method to find sufficient input subsets on high-dimensional inputs. The sufficient input subsets discovered by Batched Gradient SIS are guaranteed to be sufficient, but may be larger than those discovered by the original exhaustive SIS algorithm. Here we find small SIS subsets with Batched Gradient SIS (Figure S15). Rather than separately masking every remaining pixel at each iteration to find the pixel whose masking least reduces $f$, we use the gradient of $f$ with respect to the input pixels $\mathbf{x}$ and mask $M$, $\nabla_M f(\mathbf{x} \odot (1 - M))$, to order pixels (via a single backward pass). Instead of masking only one pixel per iteration, we mask larger subsets of $k \geq 1$ pixels per iteration. Given $p$ input features, our Batched Gradient FindSIS procedure finds each SIS subset in $\mathcal{O}(\frac{p}{k})$ evaluations of $\nabla f$ (as opposed to $\mathcal{O}(p^2)$ evaluations of $f$ in FindSIS [4]). The complete Batched Gradient SIS algorithm is presented in Section S1.

## 3.3 Detecting Overinterpretation

We produce sparse variants of all train and test set images retaining 5% (CIFAR-10) or 10% (ImageNet) of pixels in each image. Our goal is to identify sparse pixel-subsets that contain feature patterns the model identifies as strong class-evidence as it classifies an image. We identify pixels to retain based on sorting by SIS BackSelect [4] (CIFAR-10) or our Batched Gradient BackSelect procedure (ImageNet). These backward selection (BS) pixel-subset images contain the final pixels

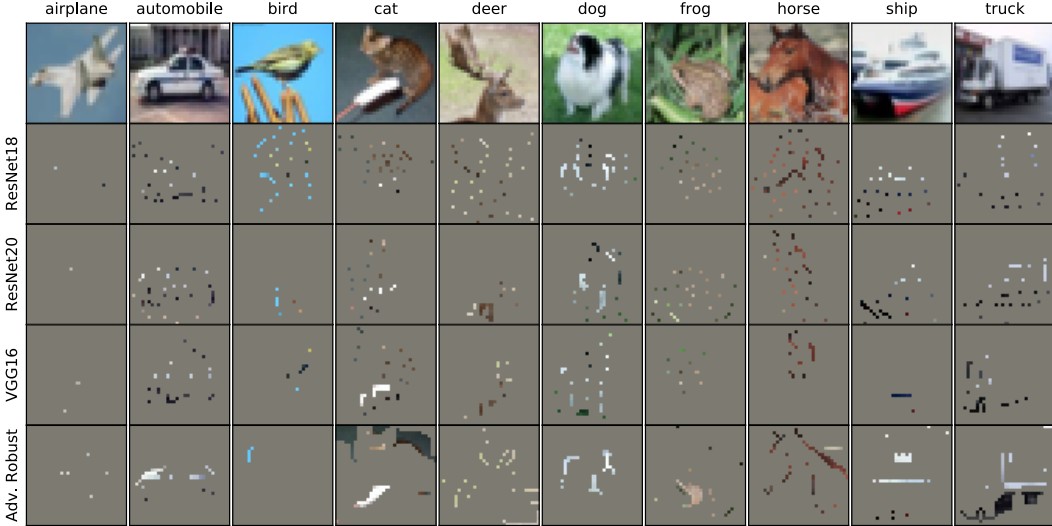

Figure 1: Sufficient input subsets (SIS) for a sample of CIFAR-10 test images (top). Each SIS image shown below is classified by the respective model with $\geq 99\%$ confidence.

(with their same RGB values as in the original images) while all other pixels' values are replaced with zero. Note that we apply backward selection to the function giving the confidence of the *predicted* class from the original model to prevent adding information about the true class for misclassified images, and we use the true labels for training/evaluating models on pixel-subsets. As backward selection is applied locally on each image, the specific pixels retained differ across images.

We train new classifiers on solely these pixel-subsets of training images and evaluate accuracy on corresponding pixel-subsets of test images to determine whether such pixel-subsets are statistically valid for generalization in the benchmark. We use the same training setup and hyperparameters (Section 3.1) without data augmentation of training images (results with data augmentation in Table S1). We consider a model to overinterpret its input when these signals can generalize to test data but lack semantic meaning (Section 3.4).

### 3.4   Human Classification Benchmark

To evaluate whether sparse pixel-subsets of images can be accurately classified by humans, we asked four participants to classify images containing various degrees of masking. We randomly sampled 100 images from the CIFAR-10 test set (10 images per class) that were correctly and confidently ($\geq 99\%$ confidence) classified by our models, and for each image, kept only 5%, 30%, or 50% of pixels as ranked by backward selection (all other pixels masked). Backward selection image subsets are sampled across our three models. Since larger subsets of pixels are by construction supersets of smaller subsets identified by the same model, we presented each batch of 100 images in order of increasing subset size and shuffled the order of images within each batch. Users were asked to classify each of the 300 images as one of the 10 classes in CIFAR-10 and were not provided training images. The same task was given to each user (and is shown in Section S5).

## 4   Results

### 4.1   CNNs Classify Images Using Spurious Features

**CIFAR-10.**   Figure 1 shows example SIS subsets (threshold 0.99) from CIFAR-10 test images (additional examples in Section S3). These SIS subset images are confidently and correctly classified by each model with $\geq 99\%$ confidence toward the predicted class. We observe these SIS subsets are highly sparse and the average SIS size at this threshold is $< 5\%$ of each image (Figure 2), suggesting these CNNs confidently classify images that appear nonsensical to humans (Section 4.3), leading to

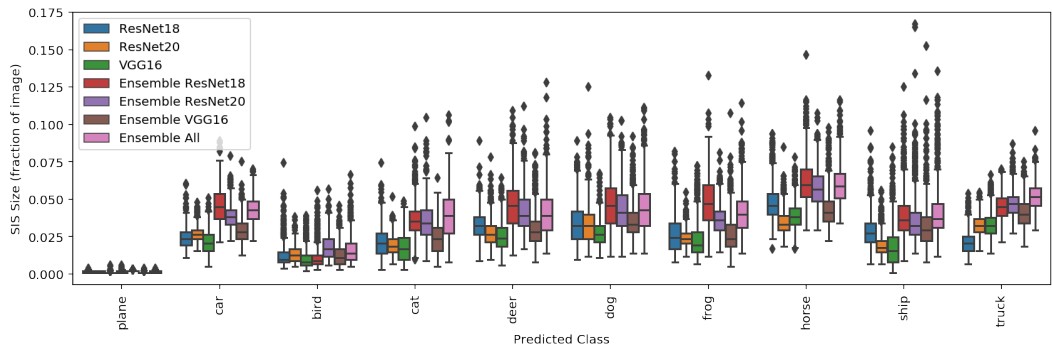

Figure 2: Distribution of SIS size per predicted class by CIFAR-10 models computed on all CIFAR-10 test set images classified with $\geq 99\%$ confidence (SIS confidence threshold 0.99).

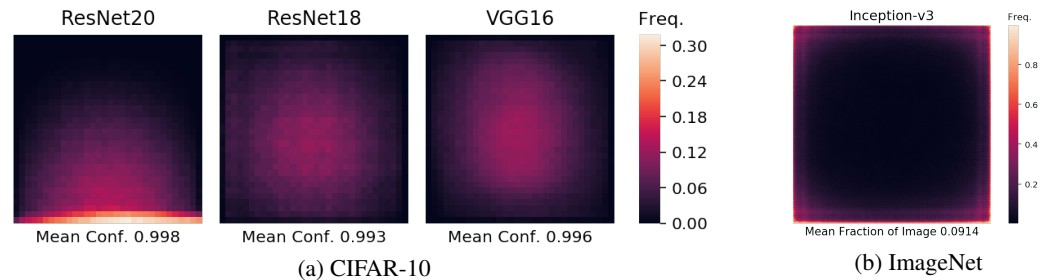

Figure 3: Heatmaps of pixel locations comprising pixel-subsets. Frequency indicates fraction of subsets containing each pixel. **(a)** 5% pixel-subsets across CIFAR-10 test set for each model. Mean confidence indicates confidence on 5% pixel-subsets. **(b)** Sufficient input subsets (confidence threshold 0.9) across ImageNet validation images from Inception v3.

concern about their robustness and generalizability. We also find that SIS size can differ significantly by predicted class (Figure 2).

We retain 5% of pixels in each image using local backward selection and mask the remaining 95% with zeros (Section 3.3) and find models trained on full images classify these pixel-subsets as accurately as full images (Table 1). Figure 3a shows the pixel locations and confidence of these 5% pixel-subsets across all CIFAR-10 test images. We found the concentration of pixels on the bottom border for ResNet20 is a result of tie-breaking during SIS backward selection (Section S4). Moreover, the CNNs are more confident on these pixels subsets than on full images: the mean drop in confidence for the predicted class between original images and these 5% subsets is $-0.035$ (std dev. $= 0.107$), $-0.016$ $(0.094)$, and $-0.012$ $(0.074)$ computed over all CIFAR-10 test images for our ResNet20, ResNet18, and VGG16 models, respectively, suggesting severe overinterpretation (negative values imply greater confidence on the 5% subsets). We find pixel-subsets chosen via backward selection are significantly more predictive than equally large pixel-subsets chosen uniformly at random from each image (Table 1).

We also find SIS subsets confidently classified by one model do not transfer to other models. For instance, 5% pixel-subsets derived from CIFAR-10 test images using one ResNet18 model (which classifies them with $94.8\%$ accuracy) are only classified with $25.8\%$, $29.2\%$, and $27.5\%$ accuracy by another ResNet18 replicate, ResNet20, and VGG16 models, respectively, suggesting there exist many different statistical patterns that a flexible model might learn to rely on, and thus CIFAR-10 image classification remains a highly underdetermined problem. Training classifiers that make predictions for the right reasons may require clever regularization strategies and architecture design to ensure models favor salient features over spurious pixel subsets.

While recent work has suggested semantics can be better captured by models that are robust to adversarial inputs that fool standard neural networks via human-imperceptible modifications to images [23, 38], we explore a wide residual network that is adversarially robust for CIFAR-10

Table 1: Accuracy of CIFAR-10 classifiers trained and evaluated on full images, 5% backward selection (BS) pixel-subsets, and 5% random pixel-subsets. Where possible, accuracy is reported as mean $\pm$ standard deviation (%) over five runs. For training on BS subsets, we run BS on all images for a single model of each type and average over five models trained on these subsets. Additional results on CIFAR-10.1 are presented in Table S4.

| Model | Train On | Evaluate On | CIFAR-10 Test Acc. | CIFAR-10-C Acc. |
|---|---|---|---|---|
| ResNet20 | Full Images | Full Images | $92.52 \pm 0.09$ | $69.44 \pm 0.52$ |
| | | 5% BS Subsets | 92.48 | 70.65 |
| | | 5% Random | $9.98 \pm 0.03$ | $10.02 \pm 0.01$ |
| | 5% BS Subsets | 5% BS Subsets | $92.49 \pm 0.02$ | $70.58 \pm 0.03$ |
| | 5% Random | 5% Random | $50.25 \pm 0.19$ | $44.04 \pm 0.33$ |
| | Input Dropout (Full) | Input Dropout (Full) | $91.02 \pm 0.25$ | $75.46 \pm 0.74$ |
| ResNet18 | Full Images | Full Images | $95.17 \pm 0.21$ | $75.08 \pm 0.20$ |
| | | 5% BS Subsets | 94.76 | 75.15 |
| | | 5% Random | $10.08 \pm 0.15$ | $10.08 \pm 0.07$ |
| | 5% BS Subsets | 5% BS Subsets | $94.96 \pm 0.04$ | $75.25 \pm 0.05$ |
| | 5% Random | 5% Random | $51.27 \pm 0.82$ | $45.24 \pm 0.45$ |
| | Input Dropout (Full) | Input Dropout (Full) | $94.15 \pm 0.26$ | $80.35 \pm 0.39$ |
| VGG16 | Full Images | Full Images | $93.69 \pm 0.12$ | $74.14 \pm 0.45$ |
| | | 5% BS Subsets | 93.27 | 73.95 |
| | | 5% Random | $10.02 \pm 0.18$ | $9.97 \pm 0.18$ |
| | 5% BS Subsets | 5% BS Subsets | $92.60 \pm 0.08$ | $73.27 \pm 0.18$ |
| | 5% Random | 5% Random | $53.66 \pm 1.96$ | $46.88 \pm 1.27$ |
| | Input Dropout (Full) | Input Dropout (Full) | $91.09 \pm 0.15$ | $80.43 \pm 0.24$ |
| Ensemble (ResNet18) | Full Images | Full Images | 96.07 | 77.00 |
| | | 5% Random | 9.98 | 10.01 |

classification [23] and find evidence of overinterpretation (Figure 1). This finding suggests adversarial robustness alone does not prevent models from overinterpreting spurious signals in CIFAR-10.

We also ran Batched Gradient SIS on CIFAR-10 and found edge-heavy sufficient input subsets for CIFAR-10 (Section S4). These heatmap differences are a result of the different valid equivalent sufficient input subsets found by the two SIS discovery algorithms. However, since all sufficient input subsets are validated with a model and guaranteed to be sufficient for classification at the specified threshold, the heatmaps are accurate depictions of what is sufficient for the model to classify images at the threshold. Overinterpretation is independent of the SIS algorithm used because both algorithms produce human-uninterpretable sufficient subsets as shown in the examples.

**ImageNet.** We find models trained on ImageNet images suffer from severe overinterpretation. Figure 4 shows example SIS subsets (threshold 0.9) found via Batched Gradient SIS on images confidently classified by the pre-trained Inception v3 (additional examples in Figures S12–S14). These SIS subsets appear visually nonsensical, yet the network classifies them with $\geq 90\%$ confidence. We find SIS pixels are concentrated outside of the actual object that determines the class label. For example, in the "pizza" image, the SIS is concentrated on the shape of the plate and the background table, rather than the pizza itself, suggesting the model could generalize poorly on images containing different circular items on a table. In the "giant panda" image, the SIS contains bamboo, which likely appeared in the collection of ImageNet photos for this class. In the "traffic light" and "street sign" images, the SIS consists of pixels in sky, suggesting that autonomous vehicle systems that may depend on these models should be carefully evaluated for overinterpretation pathologies.

Figure 3b shows SIS pixel locations from a random sample of 1000 ImageNet validation images. We find concentration along image borders, suggesting the model relies heavily on image backgrounds and suffers from severe overinterpretation. This is a serious problem as objects determining ImageNet

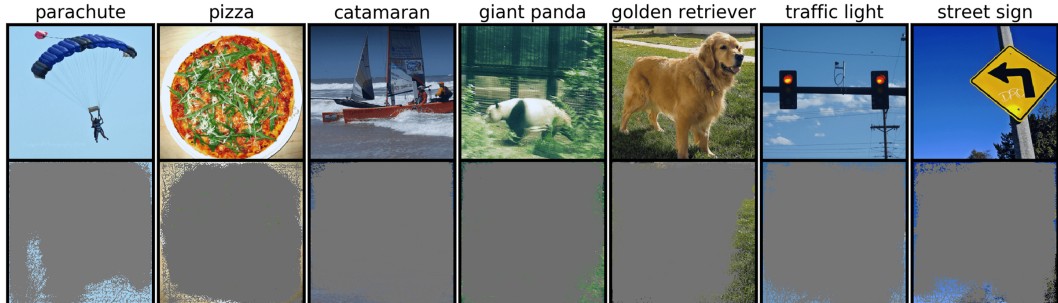

Figure 4: Sufficient input subsets (threshold 0.9) for example ImageNet validation images. The bottom row shows the corresponding images with all pixels outside of each SIS subset masked but are still classified by the Inception v3 model with $\geq 90\%$ confidence.

classes are often located near image centers, and thus this network fails to focus on salient features. We found the mean fraction of an image required for classification with $\geq 90\%$ confidence is only 0.0914, and mean SIS size differs significantly by predicted class (Figure S16).

## 4.2 Sparse Subsets are Real Statistical Patterns

The overconfidence of CNNs for image classification [39] may lead one to wonder whether the observed overconfidence on semantically meaningless SIS subsets is an artifact of calibration rather than true statistical signals in the dataset. We train models on 5% pixel-subsets of CIFAR-10 training images found via backward selection (Section 3.3). We find models trained solely on these pixel-subsets can classify corresponding test image pixel-subsets with minimal accuracy loss compared to models trained on full images (Table 1), and thus these 5% pixel-subsets are valid statistical signals in training images that generalize to the test distribution. As a baseline to the 5% pixel-subsets identified by backward selection, we create variants of all images where the 5% pixel-subsets are selected at random from each image (rather than by backward selection) and use the same random pixel-subsets for training each new model. Models trained on random subsets have significantly lower test accuracy compared to models trained on 5% pixel-subsets from backward selection (Table 1). We observe, however, that random 5% subsets of images still capture enough signal to predict roughly 5 times better than blind guessing, but do not capture nearly enough information for models to make accurate predictions.

We found that the 5% backward selection pixel-subsets did not contain model-specific features, and thus reflected valid predictive signals regardless of the model architecture employed for subset discovery. Our hypothesis was that 5% pixel-subsets discovered with one architecture would provide robust performance when used to train and evaluate a second architecture. We found this hypothesis supported for all six pairs of subset discovery and train-test architectures evaluated (Table S2). These results demonstrate that the highly sparse subsets found via backward selection offer a valid predictive signal in the CIFAR-10 benchmark exploited by models to attain high test accuracy.

We observe similar results on ImageNet. Inception v3 trained on 10% pixel-subsets of ImageNet training images achieves 71.4% top-1 accuracy (mean over 5 runs) on the corresponding pixel-subset ImageNet validation set (Table S7). Additional ImageNet results for Inception v3 and ResNet50, including training and evaluation on random pixel-subsets and pixel-subsets of different architectures, are provided in Table S7.

## 4.3 Humans Struggle to Classify Sparse Subsets

We find a strong correlation between the fraction of unmasked pixels in each image and human classification accuracy ($R^2 = 0.94$, Figure S11). Human accuracy on 5% pixel-subsets of CIFAR-10 images (mean = 19.2%, std dev = 4.8%, Table S6) is significantly lower than on original, unmasked images (roughly 94% [40]), though greater than random guessing, presumably due to correlations between labels and features such as color (e.g., blue sky suggests airplane, ship, or bird).

However, CNNs (even when trained on full images and achieve accuracy on par with human accuracy on full images) classify these sparse image subsets with very high accuracy (Table 1), indicating

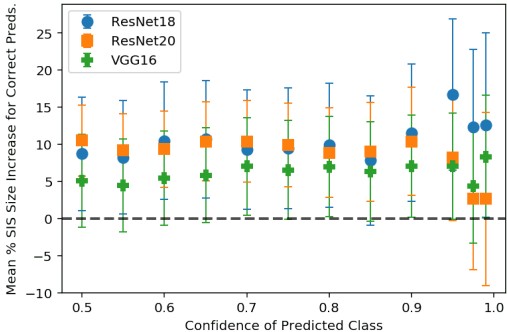
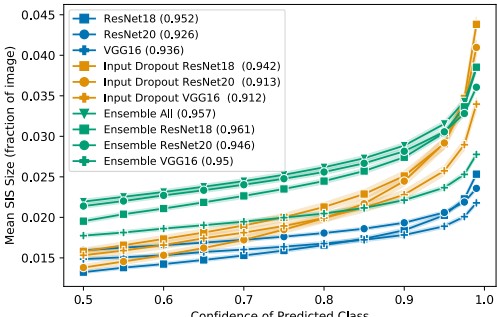

Figure 5: Percentage increase in mean SIS size of correctly classified compared to misclassified CIFAR-10 test images. Positive values indicate larger mean SIS size for correctly classified images. Error bars indicate 95% confidence interval for the difference in means.

Figure 6: Mean SIS size on CIFAR-10 test images as SIS threshold varies. SIS size indicates fraction of pixels necessary for model to make the same prediction at each confidence threshold. Model accuracies are shown in the legend. 95% confidence intervals are shaded around each mean.

benchmark images contain statistical signals that are not salient to humans. Models solely trained to minimize prediction error may thus latch onto these signals while still accurately generalizing to test data, but may behave counterintuitively when fed images from a different source that does not share these exact statistics. The strong correlation between the size of CIFAR-10 pixel-subsets and the corresponding human classification accuracy suggests larger subsets contain more semantically salient content. Thus, a model whose decisions have larger corresponding SIS subsets presumably exhibits less overinterpretation than one with smaller SIS subsets, as we investigate in Section 4.4.

## 4.4 SIS Size is Related to Model Accuracy

Given that smaller SIS contain fewer salient features according to human classifiers, models that justify their classifications based on sparse SIS subsets may be limited in terms of attainable accuracy, particularly in out-of-distribution settings. Here, we investigate the relationship between a single model's predictive accuracy and the size of the SIS subsets in which it identifies class-evidence. We draw no conclusions between models as they are uncalibrated (additional results of SIS from calibrated models are presented in Section S4). For each of our three classifiers, we compute the average SIS size increase for correctly classified images as compared to incorrectly classified images (expressed as a percentage). We find SIS subsets of correctly classified images are consistently significantly larger than those of misclassified images at all SIS confidence thresholds for both CIFAR-10 test images (Figure 5) and CIFAR-10-C OOD images (Figure S3). This is especially striking given model confidence is uniformly lower on the misclassified inputs (Figure S4). Lower confidence would normally imply a larger SIS subset at a given confidence level, as one expects fewer pixels can be masked before the model's confidence drops below the SIS threshold. Thus, we can rule out overall model confidence as an explanation of the smaller SIS of misclassified images. This result suggests the sparse SIS subsets highlighted in this paper are not just a curiosity, but may be leading to poor generalization on real images.

## 4.5 Mitigating Overinterpretation

**Ensembling.** Model ensembling is known to improve classification performance [41, 42]. As we found pixel-subset size to be strongly correlated with human pixel-subset classification accuracy (Section 4.3), our metric for measuring how much ensembling may alleviate overinterpretation is the increase in SIS subset size. We find ensembling uniformly increases test accuracy as expected but also increases the SIS size (Figure 6), hence mitigating overinterpretation.

We conjecture the cause of both the increase in the accuracy and SIS size for ensembles is the same. We observe that SIS subsets are generally not transferable from one model to another — i.e., an SIS for one model is rarely an SIS for another (Section 4.1). Thus, different models rely on different independent signals to arrive at the same prediction. An ensemble bases its prediction on multiple

such signals, increasing predictive accuracy and SIS subset size by requiring simultaneous activation of multiple independently trained feature detectors. We find SIS subsets of the ensemble are larger than the SIS of its individual members (examples in Figure S2).

**Input Dropout.** We apply input dropout [43] to both train and test images. We retain each input pixel with probability $p = 0.8$ and set the values of dropped pixels to zero. We find a small decrease in CIFAR-10 test accuracy for models regularized with input dropout though find a significant ($\sim 6\%$) increase in OOD test accuracy on CIFAR-10-C images (Table 1, Figure S5). Figure 6 shows a corresponding increase in SIS subset size for these models, suggesting input dropout applied at train and test time helps to mitigate overinterpretation. We conjecture that random dropout of input pixels disrupts spurious signals that lead to overinterpretation.

## 5   Discussion

We find that modern image classifiers overinterpret small nonsensical patterns present in popular benchmark datasets, identifying strong class evidence in the pixel-subsets that constitute these patterns. We introduced the Batched Gradient SIS method for the efficient discovery of such patterns. Despite their lack of salient features, these sparse pixel-subsets are underlying statistical signals that suffice to accurately generalize from the benchmark training data to the benchmark test data. We found that different models rationalize their predictions based on different sufficient input subsets, suggesting optimal image classification rules remain highly underdetermined by the training data. In high-stakes applications, we recommend ensembles of networks or regularization via input dropout.

Our results call into question model interpretability methods whose outputs are encouraged to align with prior human beliefs of proper classifier operating behavior [44]. Given the existence of non-salient pixel-subsets that alone suffice for correct classification, a model may solely rely on such patterns. In this case, an interpretability method that faithfully describes the model should output these nonsensical rationales, whereas interpretability methods that bias rationales toward human priors may produce results that mislead users to think their models behave as intended.

Mitigating overinterpretation and the broader task of ensuring classifiers are accurate for the right reasons remain significant challenges for ML. While we identify strategies for partially mitigating overinterpretation, additional research needs to develop ML methods that rely exclusively on well-formed interpretable inputs, and methods for creating training data that do not contain spurious signals. One alternative is to regularize CNNs by constraining the pixel attributions generated via a saliency map [45–47]. Unfortunately, such methods require a human annotator to highlight the correct pixels as an auxiliary supervision signal. Saliency maps have also been shown to provide unreliable insights into model operating behavior and must be interpreted as approximations [48]. In contrast, our SIS subsets constitute actual pathological examples that have been misconstrued by the model. An important application of our methods is the evaluation of training datasets to ensure decisions are made on interpretable rather than spurious signals. We found popular image datasets contain such spurious signals, and the resulting overinterpretation may be difficult to overcome with ML methods alone.

## Acknowledgments and Disclosure of Funding

This work was supported by Schmidt Futures and the National Institutes of Health [R01CA218094].

## Author Contributions

All authors contributed to conceptualization, methodology, formal analysis, and writing. BC led execution of the experiments.

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
