# Supplementary Material:
# Overinterpretation reveals image classification model pathologies

**Brandon Carter**
MIT CSAIL
bcarter@csail.mit.edu

**Siddhartha Jain**
MIT CSAIL

**Jonas Mueller**
Amazon Web Services

**David Gifford**
MIT CSAIL
gifford@mit.edu

## Contents

# S1  Details of Batched Gradient SIS Algorithm

It is computationally infeasible to scale the original backward selection procedure of SIS [4] to ImageNet. As each ImageNet image contains $299 \times 299 = 89401$ pixels, running backward selection to find one SIS for an image would require $\sim 4$ billion forward passes through the network. Here we introduce a more efficient gradient-based approximation to the original SIS procedure (via **Batched Gradient SIScollection**, **Batched Gradient BackSelect**, and **Batched Gradient FindSIS**) that allows us to find SIS on larger ImageNet images in a reasonable time. The **Batched Gradient SIScollection** procedure described below identifies a complete collection of disjoint masks for an input $\mathbf{x}$, where each mask $M$ specifies a pixel-subset of the input $\mathbf{x}_S = \mathbf{x} \odot (1 - M)$ such that $f(\mathbf{x}_S \geq \tau)$. Here $f$ outputs the probability assigned by the network to its predicted class (i.e., its confidence).

The idea behind our approximation algorithm is two-fold: (1) Instead of separately masking every remaining pixel to find the least critical pixel (whose masking least reduces the confidence in the network's prediction), we use the *gradient* with respect to the mask as a means of ordering. (2) Instead of masking just 1 pixel per iteration, we mask larger subsets of $k \geq 1$ pixels per iteration. More formally, let $\mathbf{x}$ be an image of dimensions $H \times W \times C$ where $H$ is the height, $W$ the width, and $C$ the channel. Let $f(\mathbf{x})$ be the network's confidence on image $\mathbf{x}$ and $\tau$ the target SIS confidence threshold. Recall that we only compute SIS for images where $f(\mathbf{x}) \geq \tau$. Let $M$ be the mask with dimensions $H \times W$ with 0 indicating an unmasked feature (pixel) and 1 indicating a masked feature. We initialize $M$ as all 0s (all features unmasked). At iteration $i$, we compute the gradient of $f$ with respect to the input pixels and mask $\nabla M = \nabla_M f(\mathbf{x} \odot (1 - M))$. Here $M$ is the current mask updated after each iteration. In each iteration, we find the block of $k$ features to mask, $G^*$, chosen in descending order by value of entries in $\nabla M$. The mask is updated after each iteration by masking this block of $k$ features until all features have been masked. Given $p$ input features, our **Batched Gradient SIScollection** procedure returns $j$ sufficient input subsets in $\mathcal{O}(\frac{p}{k} \cdot j)$ evaluations of $\nabla f$ (as opposed to $\mathcal{O}(p^2 j)$ evaluations of $f$ in the original SIS procedure [4]).

We use $k = 100$ in this paper, which allows us to find one SIS for each of 32 ImageNet images (i.e., a mini-batch) in $\sim$1-2 minutes using **Batched Gradient FindSIS**. Note that while our algorithm is an approximate procedure, the pixel-subsets produced are real sufficient input subsets, i.e., they always satisfy $f(\mathbf{x}_S \geq \tau)$. For CIFAR-10 images (which are smaller in size), we use the original SIS procedure from [4]. For both datasets, we treat all channels of each pixel as a single feature.

**Algorithm 1: Batched Gradient SIScollection**

**Input:** function $f$, input $\mathbf{x}$, threshold $\tau$, batch size $k$ (number of pixels)
$M = \mathbf{0}$
**for** $j = 1, 2, \ldots$ **do**
  $R =$ **Batched Gradient BackSelect**$(f, \mathbf{x}, M, k)$
  $M_j =$ **Batched Gradient FindSIS**$(f, \mathbf{x}, \tau, R)$
  $M \leftarrow M + M_j$
  **if** $f(\mathbf{x} \odot (1 - M)) < \tau$ **then**
    **return** $M_1, \ldots, M_{j-1}$
  **end if**
**end for**

---

**Algorithm 2: Batched Gradient BackSelect**

**Input:** function $f$, input $\mathbf{x}$, mask $M$, batch size $k$ (number of pixels)
$R =$ empty stack
**while** $M \neq \mathbf{1}$ **do**
  $G^* = \text{Top}_k \left( \nabla_M f(\mathbf{x} \odot (1 - M)) \right)$
  Update $M \leftarrow M + G^*$
  Push $G^*$ onto top of $R$
**end while**
**return** $R$

---

**Algorithm 3: Batched Gradient FindSIS**

**Input:** function $f$, input $\mathbf{x}$, threshold $\tau$, stack $R$
$M = \mathbf{1}$
**while** $f(\mathbf{x} \odot (1 - M)) < \tau$ **do**
  Pop $G$ from top of $R$
  Update $M \leftarrow M - G$
**end while**
**if** $f(\mathbf{x} \odot (1 - M)) \geq \tau$ **then**
  **return** $M$
**else**
  **return** *None*
**end if**

## S2  Model Implementation and Training Details

### CIFAR-10 Models

We first describe the implementation and training details for the CIFAR-10 models used in this paper (Section 3.1). The ResNet20 architecture [31] has 16 initial filters and a total of 0.27M parameters. ResNet18 [32] has 64 initial filters and contains 11.2M parameters. The VGG16 architecture [33] uses batch normalization and contains 14.7M parameters.

All models are trained for 200 epochs with a batch size of 128. We minimize cross-entropy via SGD with Nesterov momentum [34] using momentum of 0.9 and weight decay of 5e-4. The learning rate is initialized as 0.1 and is reduced by a factor of 5 after epochs 60, 120, and 160. Datasets are normalized using per-channel mean and standard deviation, and we use standard data augmentation strategies consisting of random crops and horizontal flips [32].

The adversarially robust model we evaluated is the `adv_trained` model of Madry et al. [23], available on GitHub[1].

To apply the SIS procedure to CIFAR-10 images, we use an implementation available on GitHub[2]. For confidently classified images on which we run SIS, we find one sufficient input subset per image using the FindSIS procedure. When masking pixels, we mask all channels of each pixel as a single feature.

### ImageNet Models

For finding SIS, we use pre-trained models (Inception v3 [36] and ResNet50 [31]) provided by PyTorch [37] in the torchvision package (PyTorch version 1.4.0, torchvision version 0.5.0).

When training new ImageNet classifiers, we adopt model implementations and training scripts from PyTorch [37], obtained from GitHub[3]. Models are trained for 90 epochs using batch size 256 (Inception-v3) or 512 (ResNet50). We minimize cross-entropy via SGD using momentum of 0.9 and weight decay of 1e-4. The learning rate is initialized as 0.1 and reduced by a factor of 10 every 30 epochs. Datasets are normalized using per-channel mean and standard deviation. For Inception v3, images are cropped to 299 x 299 pixels. For ResNet50, images are cropped to 224 x 224. When training Inception v3, we define the model using the `aux_logits=False` argument. We do not use data augmentation when training models on pixel-subsets of images.

### Hardware Details

Each CIFAR-10 model is trained on 1 NVIDIA GeForce RTX 2080 Ti GPU. Once models are trained, SIS are computed across multiple GPUs (by parallelizing over individual images). Each SIS (for 1 CIFAR-10 image) takes roughly 30-60 seconds to compute (depending on the model architecture).

ImageNet models are trained on 2–3 NVIDIA Titan RTX GPUs. For finding SIS from pre-trained ImageNet models, we run Batched Gradient BackSelect for batches of 32 images across 10 NVIDIA GeForce RTX 2080 Ti GPUs, which takes roughly 1-2 minutes per batch (details in Section S1).

---

[1] https://github.com/MadryLab/cifar10_challenge
[2] https://github.com/google-research/google-research/blob/master/sufficient_input_subsets/sis.py
[3] https://github.com/pytorch/examples/blob/master/imagenet/main.py

## S3 Additional Examples of CIFAR-10 Sufficient Input Subsets

### S3.1 SIS of Individual Networks

Figure S1 shows a sample of SIS for each of our three architectures. These images were randomly sampled among all CIFAR-10 test images confidently (confidence $\geq 0.99$) predicted to belong to the class written on the left. Out of 10000 CIFAR-10 test images, 8596 were predicted with $\geq 99\%$ confidence by ResNet18 (7829 by ResNet20, 9048 by VGG16). SIS are computed under a threshold of 0.99, so all images shown in this figure are classified with probability $\geq 99\%$ confidence as belonging to the listed class.

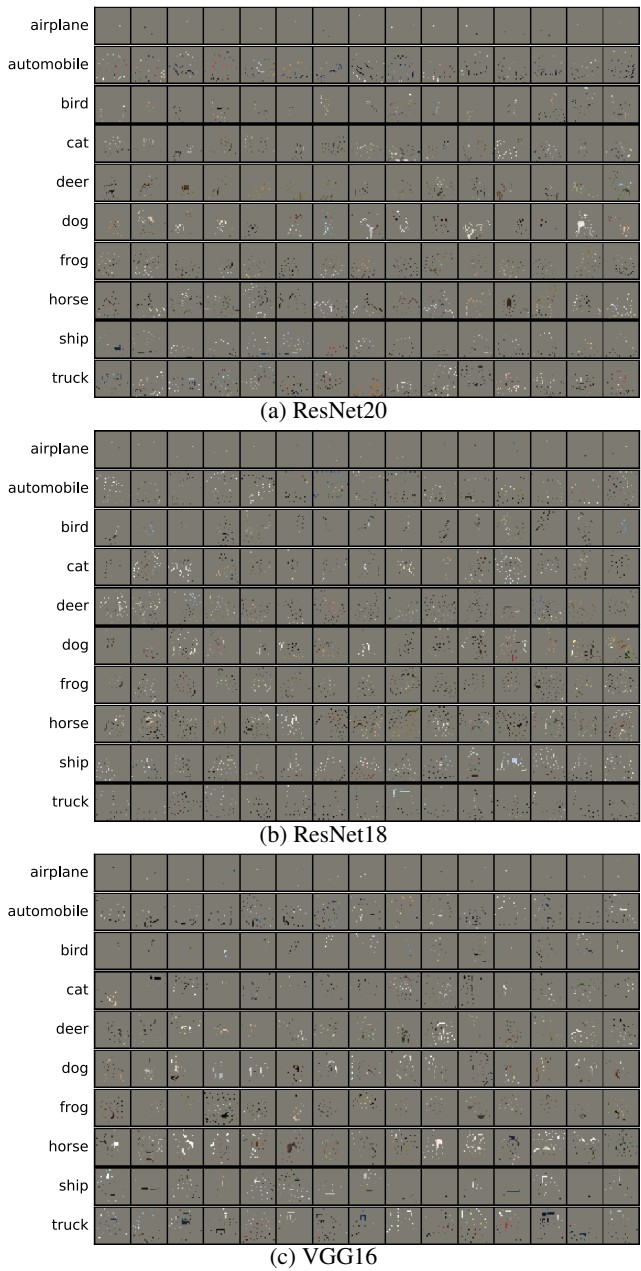

Figure S1: Examples of SIS (threshold 0.99) on random sample of CIFAR-10 test images (15 per class, different random sample for each architecture). All images shown here are predicted to belong to the listed class with $\geq 99\%$ confidence.

## S3.2 Ensemble Sufficient Input Subsets

Figure S2 shows examples of SIS from one of our model ensembles (a homogeneous ensemble of ResNet18 networks, see Section 3.1), along with corresponding SIS for the same image from each of the five member networks in the ensemble. We use a SIS threshold of 0.99, so all images are classified with $\geq 99\%$ confidence. These examples highlight how the ensemble SIS are larger and draw class-evidence from the individual members' SIS.

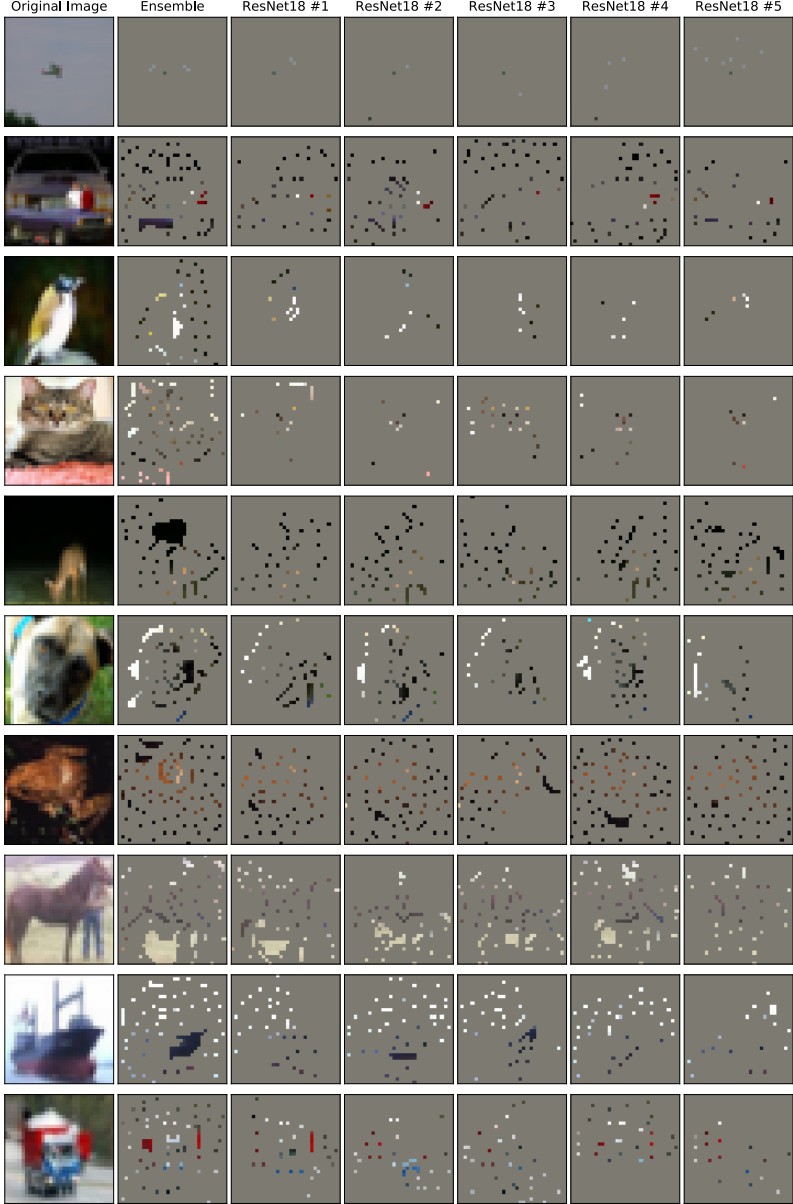

Figure S2: Examples of SIS (threshold 0.99) from the ResNet18 homogeneous ensemble (Section 3.1) and its member models. Each row shows original CIFAR-10 image (left), followed by SIS from the ensemble (second column) and the SIS from each of its 5 member networks (remaining columns). Each image shown is classified with $\geq 99\%$ confidence by its respective network.

## S4 Additional Results on CIFAR-10

### S4.1 Training on Pixel-Subsets With Data Augmentation

Table S1 presents results similar to those in Section 4.2 and Table 1, but where models are trained on 5% pixel-subsets with data augmentation (as described in Section S2). We find training without data augmentation slightly improves accuracy when training classifiers on 5% pixel-subsets of CIFAR-10.

Table S1: Accuracy of CIFAR-10 classifiers trained and evaluated on full images, 5% backward selection (BS) pixel-subsets, and 5% random pixel-subsets *with* data augmentation (+). Accuracy is reported as mean $\pm$ standard deviation (%) over five runs.

| Model | Train On | Evaluate On | CIFAR-10 Test Acc. | CIFAR-10-C Acc. |
|---|---|---|---|---|
| ResNet20 | 5% BS Subsets (+) | 5% BS Subsets | $92.26 \pm 0.01$ | $70.21 \pm 0.14$ |
| | 5% Random (+) | 5% Random | $48.87 \pm 0.41$ | $42.66 \pm 0.15$ |
| ResNet18 | 5% BS Subsets (+) | 5% BS Subsets | $94.51 \pm 0.38$ | $74.91 \pm 0.41$ |
| | 5% Random (+) | 5% Random | $49.03 \pm 0.92$ | $42.97 \pm 0.82$ |
| VGG16 | 5% BS Subsets (+) | 5% BS Subsets | $91.17 \pm 0.04$ | $71.82 \pm 0.13$ |
| | 5% Random (+) | 5% Random | $51.32 \pm 1.35$ | $44.56 \pm 0.96$ |

### S4.2 Training on Pixel-Subsets With Different Architectures

Table S2 presents results of training and evaluating models on 5% pixel-subsets drawn from different architectures. Models were trained without data augmentation on subsets from one replicate of each base architecture. We find accuracy from training and evaluating a model on 5% pixel-subsets of images derived from a different architecture is commensurate with accuracy of training and evaluating a new model of the same type on those subsets (Table 1).

Table S2: Accuracy of CIFAR-10 classifiers trained and evaluated on 5% backward selection (BS) pixel-subsets from different architectures. Accuracy is reported as mean $\pm$ standard deviation (%) over five runs.

| 5% Subsets from Model | Model Trained | CIFAR-10 Test Acc. | CIFAR-10-C Acc. |
|---|---|---|---|
| ResNet20 | ResNet18 | $92.53 \pm 0.02$ | $70.56 \pm 0.04$ |
| | VGG16 | $92.47 \pm 0.02$ | $70.42 \pm 0.14$ |
| ResNet18 | ResNet20 | $94.88 \pm 0.03$ | $75.14 \pm 0.10$ |
| | VGG16 | $94.88 \pm 0.05$ | $75.13 \pm 0.09$ |
| VGG16 | ResNet20 | $92.05 \pm 0.14$ | $73.01 \pm 0.08$ |
| | ResNet18 | $92.57 \pm 0.10$ | $73.33 \pm 0.21$ |

### S4.3 Additional Results for Models Trained on Pixel-Subsets

Table S3 presents results of models trained on 5% backward selection (BS) or random pixel-subsets of CIFAR-10 training images, evaluated on full (original) CIFAR-10 test images. While accuracies are generally significantly higher than random guessing, we note that full images are highly out-of-distribution for a model trained on images with only 5% pixel-subsets and hence such a model cannot properly generalize to full images. Further, the model trained on 5% images may not rely on the same features as the model trained on full images as it is trained on a substantially different training set.

Table S3: Accuracy of CIFAR-10 classifiers trained on 5% backward selection (BS) or random pixel-subsets with (+) and without (−) data augmentation. Accuracy is reported as mean ± standard deviation (%) over five runs.

| Model | Train On | Evaluate On | CIFAR-10 Test Acc. | CIFAR-10-C Acc. |
|---|---|---|---|---|
| ResNet20 | 5% BS Subsets (−) | Full Images | $21.02 \pm 1.57$ | $17.50 \pm 1.15$ |
| | 5% Random (−) | Full Images | $38.66 \pm 3.31$ | $36.40 \pm 2.73$ |
| | 5% BS Subsets (+) | Full Images | $10.87 \pm 1.50$ | $10.75 \pm 1.32$ |
| | 5% Random (+) | Full Images | $37.08 \pm 3.51$ | $33.78 \pm 2.81$ |
| ResNet18 | 5% BS Subsets (−) | Full Images | $20.86 \pm 2.74$ | $18.20 \pm 1.43$ |
| | 5% Random (−) | Full Images | $26.05 \pm 7.59$ | $25.03 \pm 6.41$ |
| | 5% BS Subsets (+) | Full Images | $11.83 \pm 1.74$ | $11.48 \pm 1.15$ |
| | 5% Random (+) | Full Images | $20.98 \pm 4.61$ | $20.35 \pm 3.56$ |
| VGG16 | 5% BS Subsets (−) | Full Images | $41.63 \pm 3.55$ | $30.34 \pm 1.97$ |
| | 5% Random (−) | Full Images | $25.73 \pm 6.08$ | $23.56 \pm 4.39$ |
| | 5% BS Subsets (+) | Full Images | $14.32 \pm 3.40$ | $13.22 \pm 2.01$ |
| | 5% Random (+) | Full Images | $27.58 \pm 3.96$ | $24.92 \pm 3.10$ |

## S4.4 Additional Results for SIS Size and Model Accuracy

Figure S3 shows percentage increase in mean SIS size for correctly classified images compared to misclassified images from the CIFAR-10-C dataset.

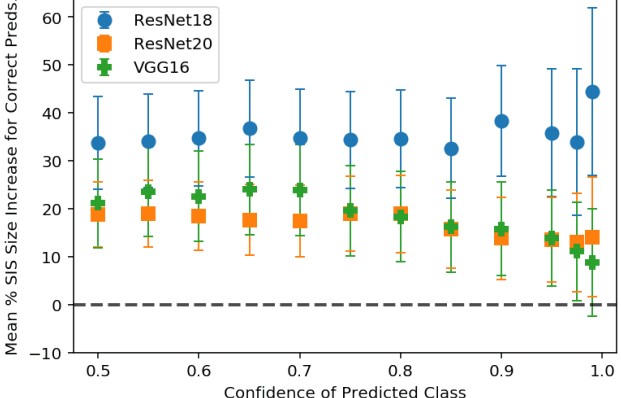

Figure S3: Percentage increase in mean SIS size of correctly classified images compared to misclassified images from a random sample of CIFAR-10-C test set. Positive values indicate larger mean SIS size for correctly classified images. Error bars indicate 95% confidence interval for the difference in means.

Figure S4 shows the mean confidence of each group of correctly and incorrectly classified images that we consider at each confidence threshold (at each confidence threshold along the x-axis, we evaluate SIS size in Figure 5 on the set of images that originally were classified with at least that level of confidence). We find model confidence is uniformly lower on the misclassified inputs.

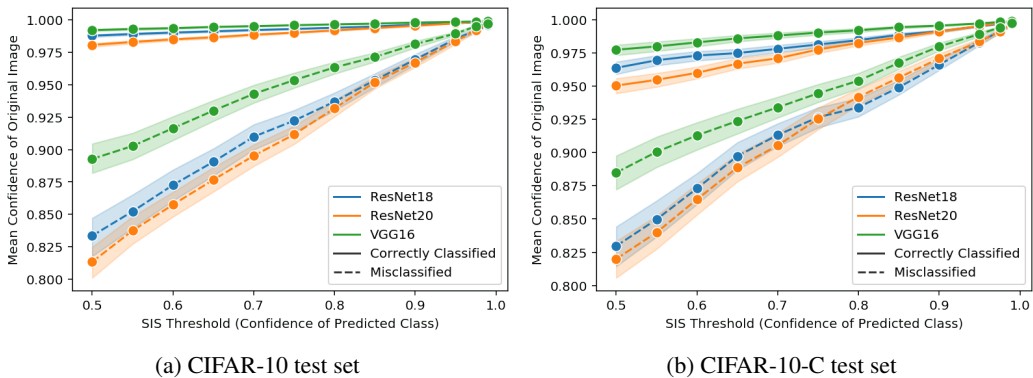

(a) CIFAR-10 test set        (b) CIFAR-10-C test set

Figure S4: Mean confidence of correctly vs. incorrectly classified images for each corresponding SIS threshold we evaluate in Figure 5 across the (a) CIFAR-10 test set and (b) our random sample of the CIFAR-10-C test set. Shaded region indicates 95% confidence interval.

## S4.5 Additional Results for Input Dropout

Figure S5 shows the accuracy improvement on each individual corruption of the CIFAR-10-C out-of-distribution test set for models trained with input dropout (Section 4.5) compared to original models.

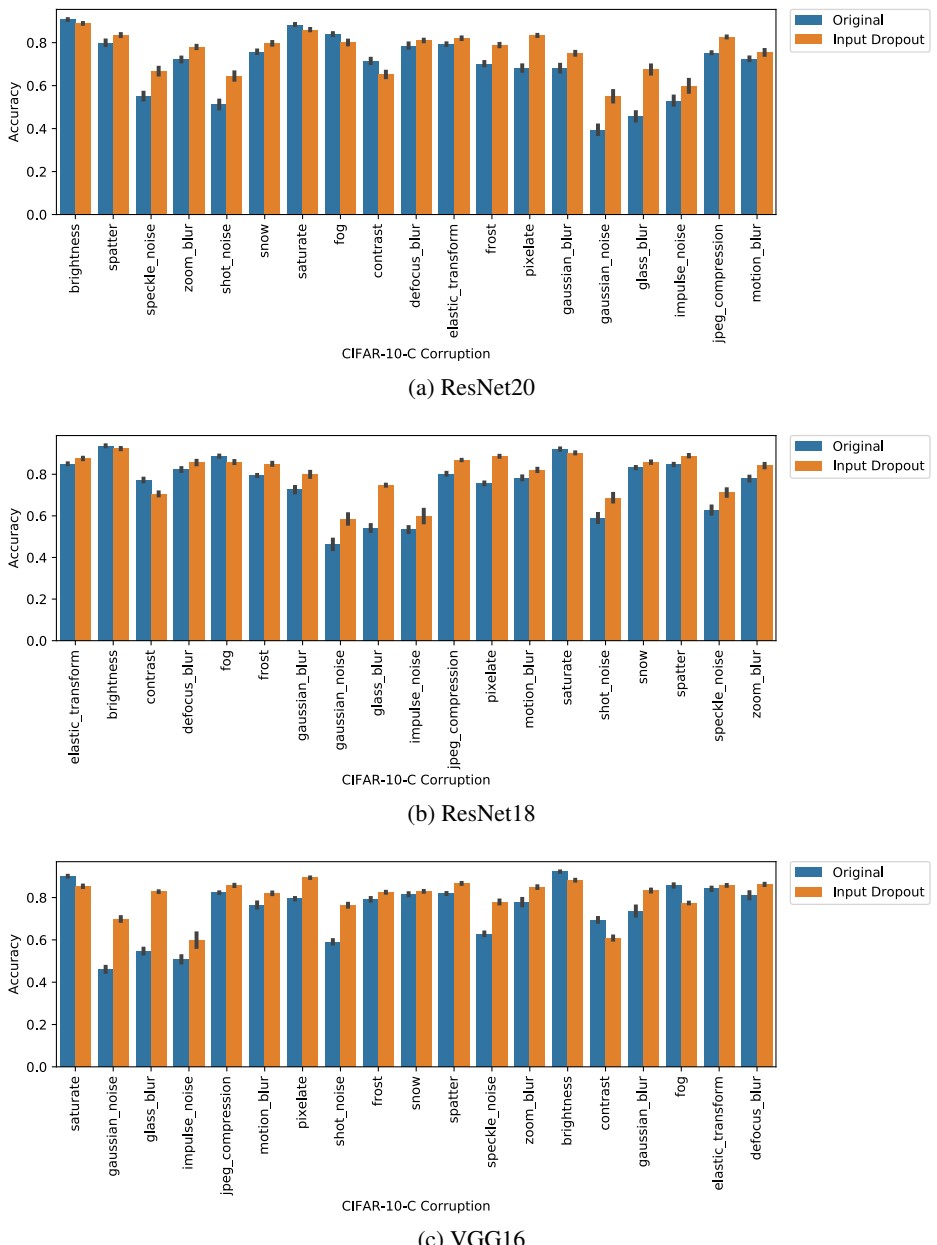

(a) ResNet20

(b) ResNet18

(c) VGG16

Figure S5: Accuracy on individual corruptions of CIFAR-10-C out-of-distribution images for original models and models trained with input dropout (Section 4.5). Accuracy is given as mean $\pm$ standard deviation over five replicate models.

## S4.6 Results on CIFAR-10.1

Table S4 reports accuracy of the models from Section 4.2 computed on the CIFAR-10.1 v6 dataset [30], which contains 2000 class-balanced images drawn from the Tiny Images repository [49] in a similar fashion to that of CIFAR-10, though Recht et al. [30] found a large drop in classification accuracy on these images.

Table S4: Accuracy of CIFAR-10 classifiers trained and evaluated on full images, 5% backward selection (BS) pixel-subsets, and 5% random pixel-subsets reported on CIFAR-10.1 v6 dataset (evaluating models from Section 4.2 that were trained on full images or 5% subsets of the CIFAR-10 train set). Where possible, accuracy is reported as mean $\pm$ standard deviation (%) over five runs. For training on BS subsets, we run BS on all images for a single model of each type and average over five models trained on these subsets.

| Model | Train On | Evaluate On | CIFAR-10.1 Acc. |
|---|---|---|---|
| ResNet20 | Full Images | Full Images | $83.98 \pm 0.68$ |
| | | 5% BS Subsets | $82.80$ |
| | | 5% Random | $10.00 \pm 0.00$ |
| | 5% BS Subsets | 5% BS Subsets | $82.56 \pm 0.07$ |
| | 5% Random | 5% Random | $39.78 \pm 1.27$ |
| | Input Dropout (Full) | Input Dropout (Full) | $81.88 \pm 0.44$ |
| ResNet18 | Full Images | Full Images | $88.89 \pm 0.45$ |
| | | 5% BS Subsets | $89.35$ |
| | | 5% Random | $10.06 \pm 0.11$ |
| | 5% BS Subsets | 5% BS Subsets | $89.49 \pm 0.04$ |
| | 5% Random | 5% Random | $39.45 \pm 1.02$ |
| | Input Dropout (Full) | Input Dropout (Full) | $86.28 \pm 0.33$ |
| VGG16 | Full Images | Full Images | $86.23 \pm 0.79$ |
| | | 5% BS Subsets | $86.45$ |
| | | 5% Random | $9.78 \pm 0.26$ |
| | 5% BS Subsets | 5% BS Subsets | $85.61 \pm 0.19$ |
| | 5% Random | 5% Random | $40.98 \pm 1.27$ |
| | Input Dropout (Full) | Input Dropout (Full) | $81.00 \pm 0.65$ |
| Ensemble (ResNet18) | Full Images | Full Images | $90.30$ |
| | | 5% Random | $10.05$ |

## S4.7 SIS and Calibrated Models

We calibrated one model of each architecture class after training using Temperature Scaling [39] based on an implementation available on GitHub[4]. The CIFAR-10 test set was randomly split into a 5k validation set (for optimization of the temperature parameter) and a 5k held-out test set (for final evaluation of ECE). Table S5 shows the Expected Calibration Error (ECE) of each model on held-out test images before and after calibration, as well as mean SIS size using confidence threshold 0.99 computed on the entire CIFAR-10 test set. We find that while the mean SIS size (for test images that the re-calibrated model can classify with $\geq 99\%$ confidence) does increase slightly, the resulting SIS subsets are still semantically meaningless and far below the threshold of SIS size where humans can meaningfully start to classify CIFAR images with any degree of accuracy (Figure S6). We note that one of the key findings of our paper is that even when we compute SIS subsets from uncalibrated models, those subsets still contain enough signal for training entirely new classifiers that can generalize as well to the corresponding test subsets (Section 4.2).

---

[4] https://github.com/gpleiss/temperature_scaling

Table S5: Results of model calibration by temperature scaling. Expected Calibration Error (ECE) is computed on a held-out set of 5k CIFAR-10 test images. SIS are computed using a threshold of 0.99 on all CIFAR-10 test images classified with $\geq 99\%$ confidence (and corresponding number of such images listed). SIS size is given as mean $\pm$ standard deviation.

| Model | ECE (%) | SIS Size (% of Image) | Num. Images Pred. $\geq 0.99$ |
|---|---|---|---|
| ResNet20 Uncalibrated | 3.91 | $2.36 \pm 1.21$ | 7829 |
| ResNet20 Calibrated | 0.91 | $2.94 \pm 1.39$ | 5805 |
| ResNet18 Uncalibrated | 2.49 | $2.53 \pm 1.53$ | 8596 |
| ResNet18 Calibrated | 1.00 | $3.54 \pm 1.94$ | 5934 |
| VGG16 Uncalibrated | 4.95 | $2.18 \pm 1.37$ | 9048 |
| VGG16 Calibrated | 1.56 | $8.26 \pm 2.86$ | 23 |

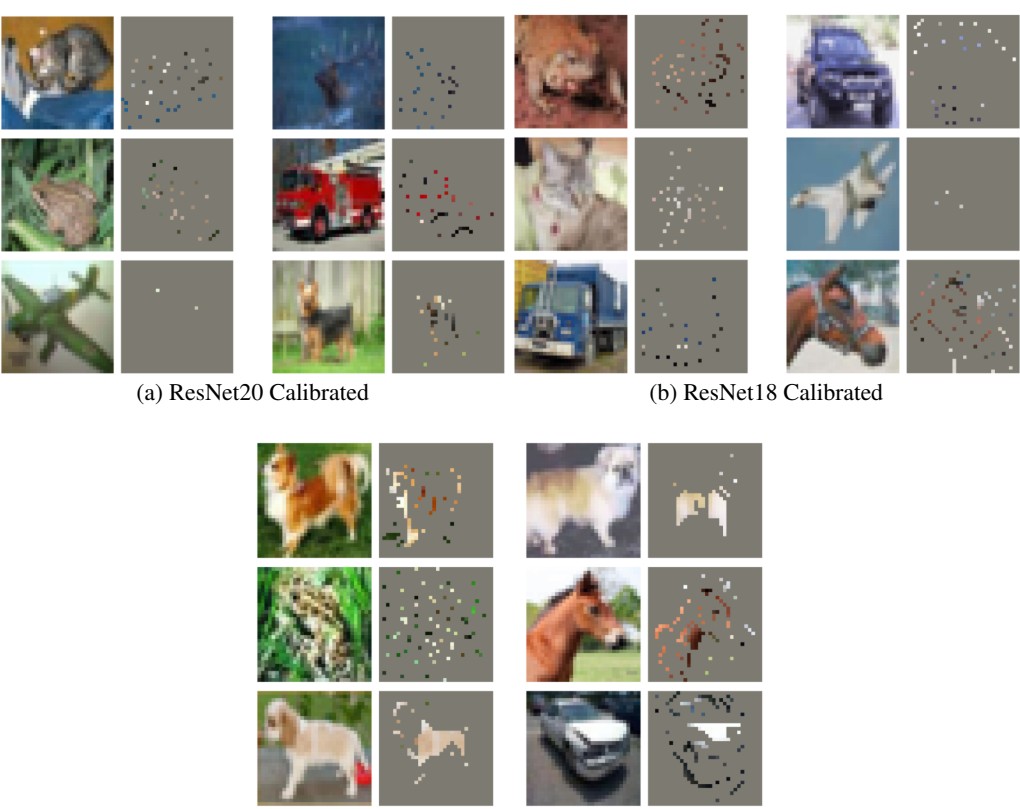

(a) ResNet20 Calibrated        (b) ResNet18 Calibrated

(c) VGG16 Calibrated

Figure S6: Examples of SIS (threshold 0.99) on sample of CIFAR-10 test images from calibrated models. All images shown are predicted to belong to the listed class with $\geq 99\%$ confidence.

## S4.8   SIS with Random Tie-breaking

We suspect the concentration of pixels on the bottom border for ResNet20 (Figure 3a) is a result of tie-breaking during backward selection of the SIS procedure. To explore this hypothesis, we modified the tie-breaking procedure to randomly (rather than deterministically) break ties during SIS backward selection by adding random Gaussian noise ($\mu = 0$, $\sigma^2 = 1e-12$) to the model's outputs for each remaining masked pixel at each iteration of backward selection. For each image in a sample of 1000 CIFAR-10 test images, we repeated this randomization procedure three times and found the resulting heatmap of 5% backward selection pixel-subsets for ResNet20 more concentrated in the image centers rather than bottom border (Figure S7).

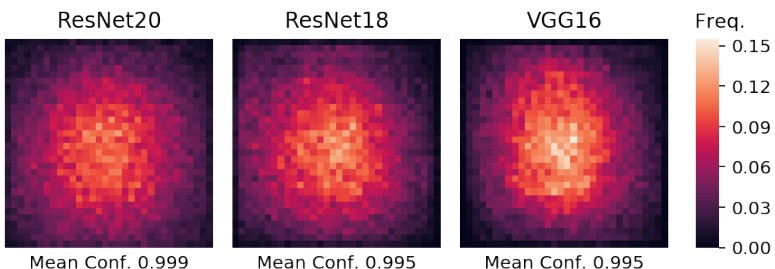

Figure S7: Heatmap of pixel locations comprising 5% backward selection pixel-subsets computed on a set of 1000 CIFAR-10 test set images with random tie-breaking during backward selection.

## S4.9 Confidence Curves for SIS Backward Selection on CIFAR-10

Figure S8 shows the predicted confidence on the remaining pixels at each step of SIS backward selection for the entire CIFAR-10 test set for each architecture trained on CIFAR-10.

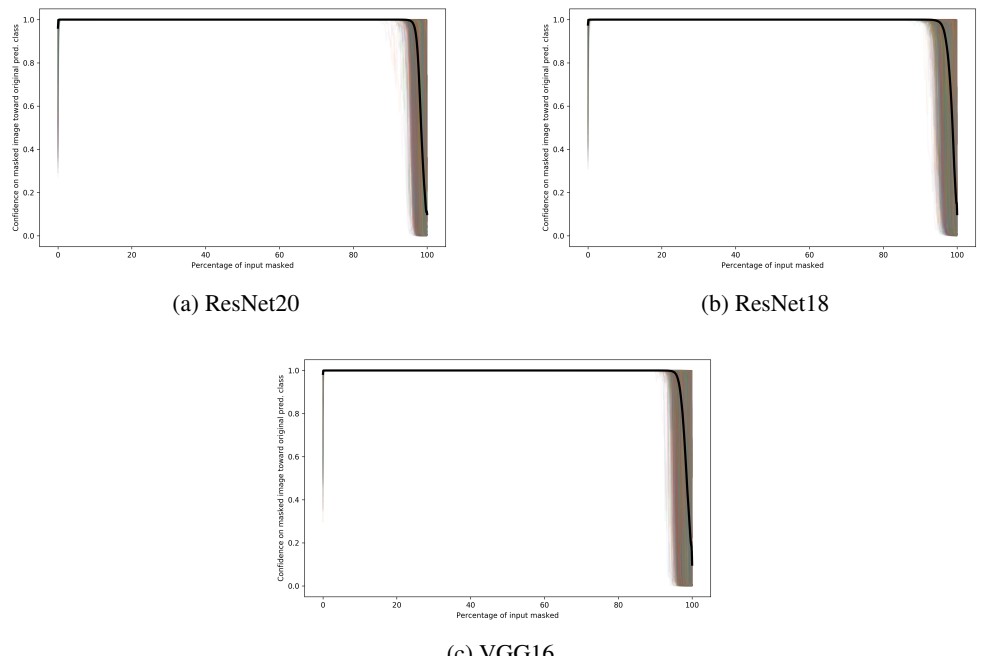

Figure S8: Prediction history on remaining (unmasked) pixels at each step of the SIS backward selection procedure for all CIFAR-10 test set images. Black line depicts mean confidence at each step.

## S4.10  Batched Gradient SIS on CIFAR-10

We also ran Batched Gradient SIS on the entire CIFAR-10 test set for ResNet18 and found Batched Gradient SIS produced edge-heavy heatmaps for CIFAR-10 (Figure S9a). For CIFAR-10, we set $k = 1$ to remove a single pixel per iteration of Batched Gradient SIS. These heatmap differences (compared to Figure 3) are a result of the different valid equivalent SIS subsets found by the two SIS discovery algorithms. However, since all SIS subsets are validated with a model and guaranteed to be sufficient for classification at the specified threshold, the heatmaps are accurate depictions of what is sufficient for the model to classify images at the threshold. Overinterpretation is independent of the SIS algorithm used because both algorithms produce human-uninterpretable sufficient subsets (Figure S9b).

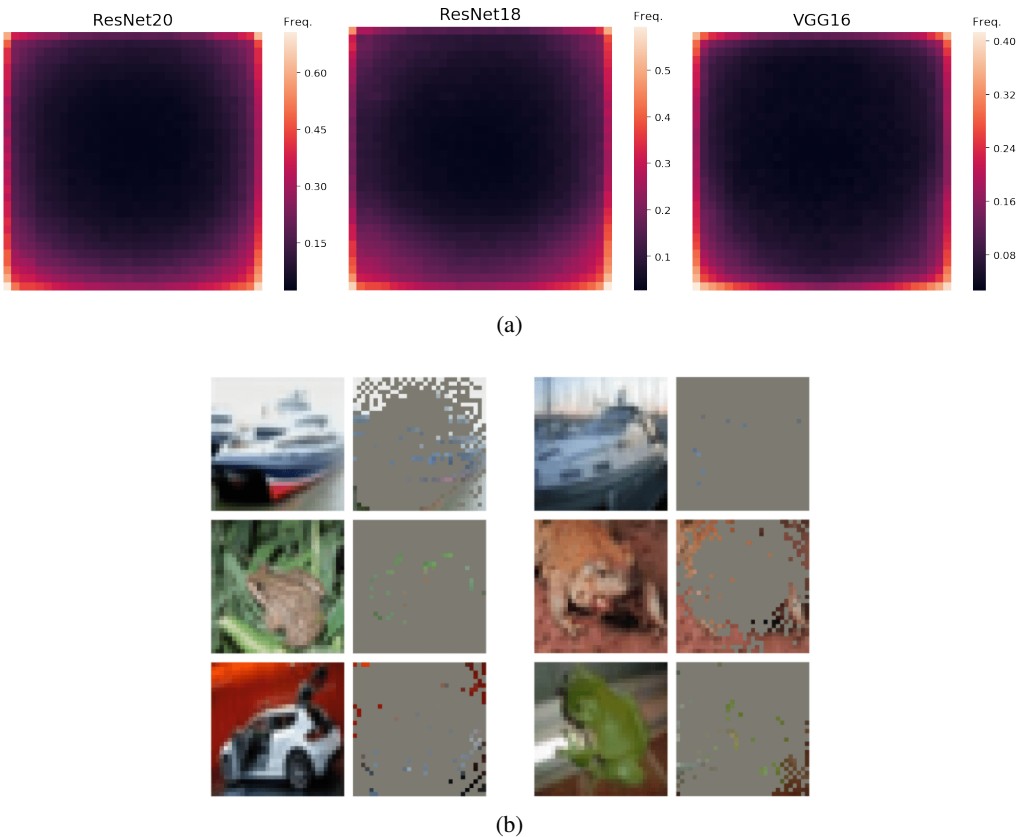

Figure S9: Results of running Batched Gradient SIS (threshold 0.99) on CIFAR-10. (a) Heatmaps of SIS pixel locations computed on entire CIFAR-10 test set for each architecture. (b) Example Batched Gradient SIS for ResNet18 (all images and SIS subsets shown are classified with $\geq$ 99% confidence).

## S5    Details of Human Classification Benchmark

Here we include additional details on our benchmark of human classification accuracy of sparse pixel-subsets (Section 3.4). Figure S10 shows all images shown to users (100 images each for 5%, 30% and 50% pixel-subsets of CIFAR-10 test images). Each set of 100 images has pixel-subsets stemming from each of the three architectures roughly equally (35 ResNet20, 35 ResNet18, 30 VGG16).[5] Figure S11 shows the correlation between human classification accuracy and pixel-subset size (accuracies shown in Table S6).

Table S6: Human classification accuracy on a sample of CIFAR-10 test image pixel-subsets of varying sparsity (see Section 3.4). Accuracies given as mean $\pm$ standard deviation.

| Fraction of Images | Human Classification Accuracy (%) |
|---|---|
| 5% | $19.2 \pm 4.8$ |
| 30% | $40.0 \pm 2.5$ |
| 50% | $68.2 \pm 3.6$ |

---

[5]The human classification benchmark was performed using pixel-subsets computed from earlier implementations of the three CNN architectures (in Keras rather than PyTorch). Figure S5 shows all pixel-subsets derived from these models that were shown to users in the human classification benchmark. ResNet20 was based on a Keras example using 16 initial filters and optimized with Adam for 200 epochs (batch size 32, initial learning rate 0.001, reduced after epochs 80, 120, 160, and 180 to 1e-4, 1e-5, 1e-6, and 5e-7, respectively). ResNet18 was based on a GitHub implementation using 64 initial filters, initial strides (1, 1), initial kernel size (3, 3), no initial pooling layer, weight decay 0.0005 and trained using SGD with Nesterov momentum 0.9 for 200 epochs (batch size 128, initial learning rate 0.1, reduced by a factor of 5 after epochs 60, 120, and 160). VGG16 was based on a GitHub implementation trained with weight decay 0.0005 and SGD with Nesterov momentum 0.9 for 250 epochs (batch size 128, initial learning rate 0.1, decayed after each epoch as $0.1 \cdot 0.5^{\lfloor \text{epoch}/20 \rfloor}$). We selected the final model checkpoint that maximized test accuracy. We found these models exhibited similar overinterpretation behavior to the final models.

- https://keras.io/examples/cifar10_resnet/
- https://github.com/keras-team/keras-contrib/blob/master/keras_contrib/applications/resnet.py
- https://github.com/geifmany/cifar-vgg/blob/e7d4bd4807d15631177a2fafabb5497d0e4be3ba/cifar10vgg.py

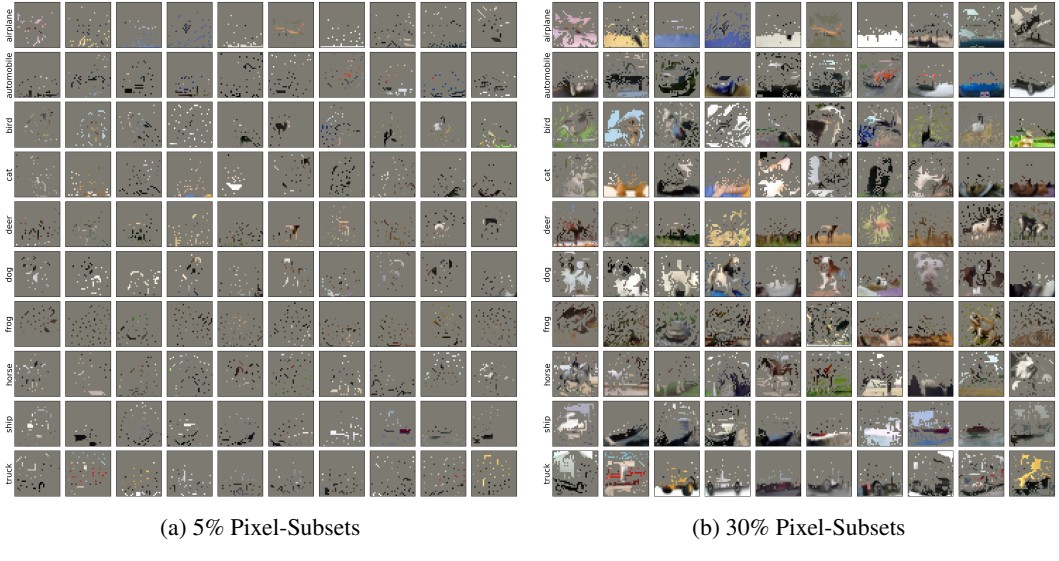

(a) 5% Pixel-Subsets

(b) 30% Pixel-Subsets

(c) 50% Pixel-Subsets

Figure S10: Pixel-subsets of CIFAR-10 test images shown to participants in our human classification benchmark (Section 3.4).

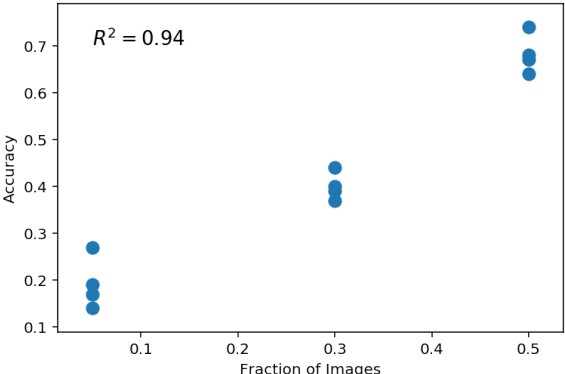

Figure S11: Human classification accuracy on a sample of CIFAR-10 test image pixel-subsets (see Section 3.4).

## S6 Additional Results of ImageNet Overinterpretation

### S6.1 Training CNNs on ImageNet Pixel-Subsets

We extracted 10% backward selection (BS) pixel-subsets by applying Batched Gradient BackSelect to all ImageNet train and validation images using pre-trained Inception v3 and ResNet50 models from PyTorch [37]. We kept the top 10% of pixels and masked the remaining 90% with zeros. We trained new models of the same type on these 10% BS pixel-subsets of ImageNet training set images (training details in Section S2) and evaluated the resulting models on the corresponding 10% pixel-subsets of ImageNet validation images. Table S7 shows a small loss in validation accuracy, suggesting these 10% pixel-subsets that are indiscernible by humans contain statistically valid signals that generalize to validation images. Models trained on 10% pixel-subsets were trained without data augmentation. As with CIFAR-10 (Section S4), we found training models on pixel-subsets with standard data augmentation techniques (random crops and horizontal flips) resulted in worse validation accuracy.

We also trained and evaluated ImageNet models on random pixel-subsets, and results are shown in Table S7. For training on random pixel-subsets, each of the five training runs was trained on different random pixel-subsets. For evaluation of pre-trained models on random subsets, each pre-trained model was evaluated on five different random random pixel-subsets. All pixels in random pixel-subsets were drawn uniformly at random, and the remaining pixels masked with zeros. We found random 10% pixel-subsets significantly less informative to pre-trained classifiers than 10% backward selection pixel-subsets from Batched Gradient SIS.

We repeated the experiment of Table S2 and found for ImageNet that 10% pixel-subsets from one architecture can also be used to train a new model of a different architecture. We trained a new DenseNet-121 model [50] on 10% BS pixel-subsets of ImageNet training images drawn from the ResNet50[6], and the DenseNet-121 was able to classify the corresponding 10% BS pixel-subsets of ImageNet validation images as accurately as the ResNet50 trained on the 10% BS pixel-subsets (Table S7).

### S6.2 Additional Examples of SIS on ImageNet

Figure S12 shows additional examples of SIS (threshold 0.9) on ImageNet validation images for the pre-trained Inception v3 found via Batched Gradient FindSIS. Figure S13 shows examples of SIS for the pre-trained ResNet50.

---

[6]we used subsets drawn from ResNet50 as the default input image size for Inception v3 is $299 \times 299$ while the default input image size for ResNet50 and DenseNet-121 is $224 \times 224$

Table S7: Accuracy of models on ImageNet validation images trained and evaluated on full images, backward selection (BS) pixel-subsets, and random pixel-subsets. Accuracy for training on 10% BS Subsets is reported as mean $\pm$ standard deviation (%) over five training runs with different random initialization. For training/evaluation on BS pixel-subsets, we run backward selection on all ImageNet images using a single pre-trained model of each type, but average over five models trained on these subsets. For training on random pixel-subsets, each of the five training runs was trained on different random pixel-subsets. For evaluation of pre-trained models on random subsets, each pre-trained model was evaluated on five different random random pixel-subsets. All pixels in random pixel-subsets were drawn uniformly at random.

| Model | Train On | Evaluate On | Top 1 Acc. | Top 5 Acc. |
|---|---|---|---|---|
| Inception v3 | Full Images (pre-trained) | Full Images | 77.21 | 93.53 |
| | | 10% BS Subsets | 73.87 | 83.43 |
| | | 15% BS Subsets | 76.15 | 84.93 |
| | | 20% BS Subsets | 76.75 | 85.40 |
| | | 10% Random | $0.75 \pm 0.02$ | $2.55 \pm 0.03$ |
| | | 15% Random | $1.51 \pm 0.03$ | $4.61 \pm 0.03$ |
| | | 20% Random | $2.83 \pm 0.03$ | $7.75 \pm 0.03$ |
| | 10% BS Subsets | 10% BS Subsets | $71.37 \pm 0.15$ | $83.73 \pm 0.10$ |
| | 10% Random | 10% Random | $64.53 \pm 0.16$ | $85.36 \pm 0.10$ |
| ResNet50 | Full Images (pre-trained) | Full Images | 76.13 | 92.86 |
| | | 10% BS Subsets | 45.14 | 64.12 |
| | | 15% BS Subsets | 61.06 | 75.26 |
| | | 20% BS Subsets | 68.35 | 79.46 |
| | | 10% Random | $0.28 \pm 0.02$ | $1.03 \pm 0.01$ |
| | | 15% Random | $0.43 \pm 0.00$ | $1.54 \pm 0.03$ |
| | | 20% Random | $0.67 \pm 0.02$ | $2.37 \pm 0.02$ |
| | 10% BS Subsets | 10% BS Subsets | $65.71 \pm 0.08$ | $80.45 \pm 0.08$ |
| | 10% Random | 10% Random | $55.70 \pm 0.24$ | $79.06 \pm 0.17$ |
| DenseNet-121 | 10% BS Subsets (from ResNet50) | 10% BS Subsets (from ResNet50) | $65.67 \pm 0.19$ | $81.30 \pm 0.10$ |

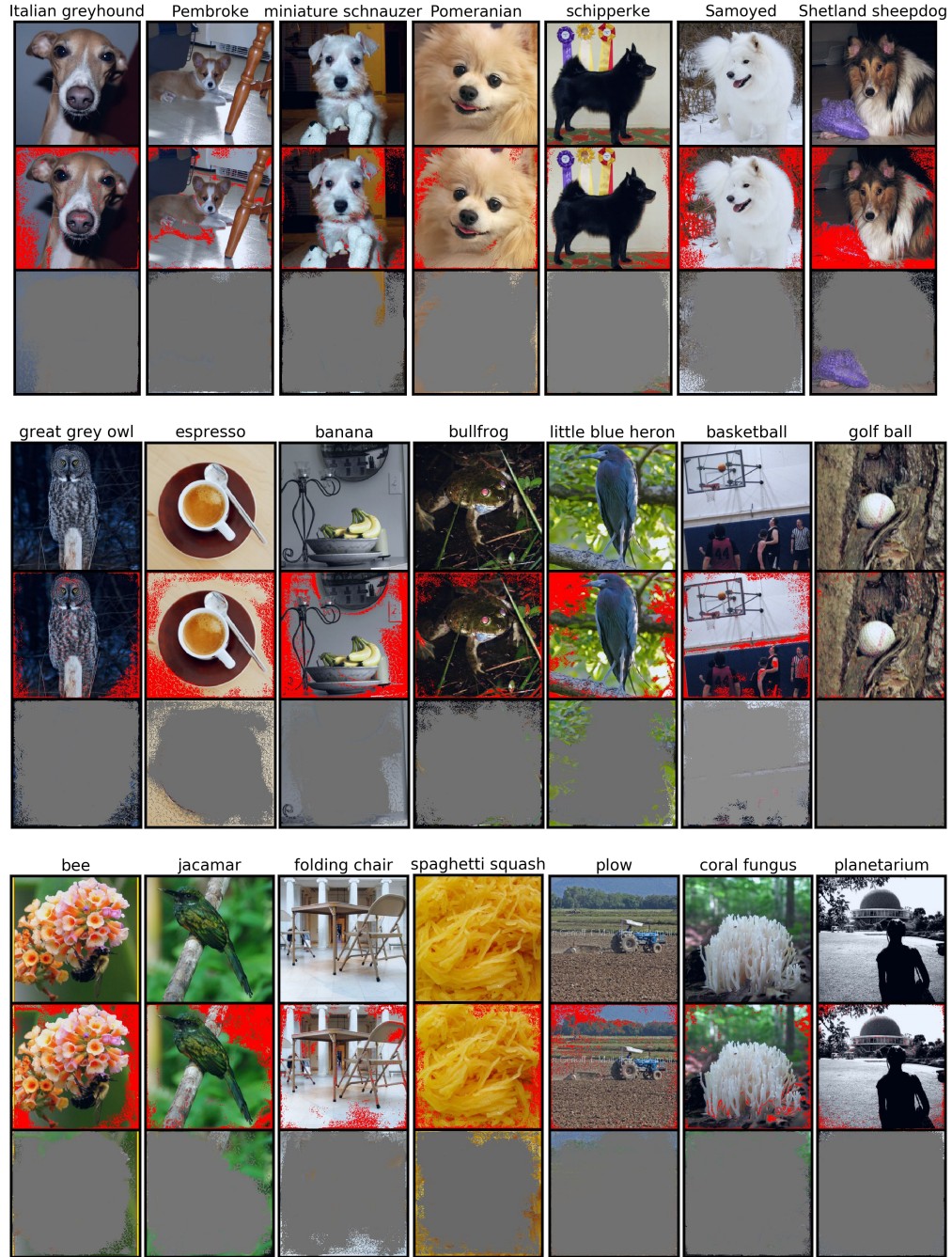

Figure S12: Example SIS (threshold 0.9) from ImageNet validation images (top row of each block) for Inception v3. The middle rows show the location of SIS pixels (red) and the bottom rows show images with all non-SIS pixels masked but are still classified by the Inception v3 model with $\geq 90\%$ confidence.

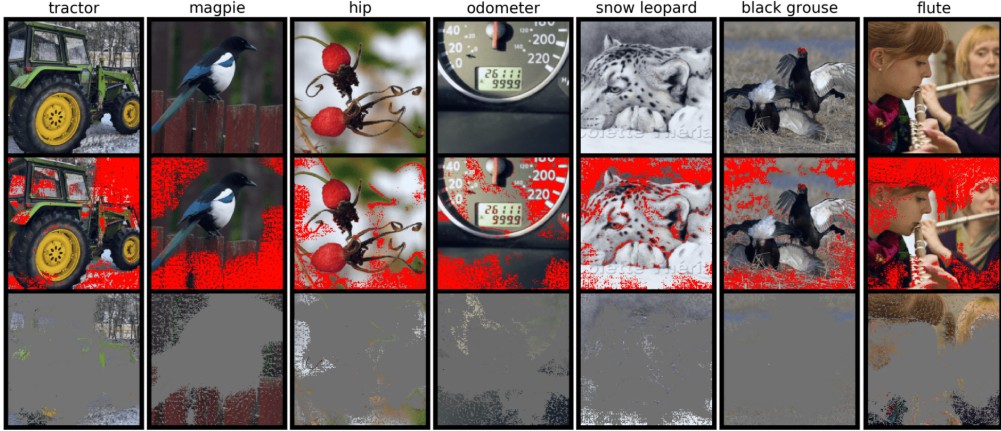

Figure S13: Example SIS (threshold 0.9) from ImageNet validation images (top row of each block) for ResNet50. The middle rows show the location of SIS pixels (red) and the bottom rows show images with all non-SIS pixels masked but are still classified by the ResNet50 model with $\geq 90\%$ confidence.

We also explored the relationship between pixel saliency and the order pixels were removed by Batched Gradient BackSelect. Surprisingly, as shown in Figure S14 for Inception v3, we found that the most salient pixels were often *eliminated first* and thus unnecessary for maintaining high predicted confidence on the remaining pixel-subsets and subsequently for training on pixel-subsets. Figure S15 shows the predicted confidence on remaining pixels at each step of the Batched Gradient BackSelect procedure for a random sample of 32 ImageNet validation images by the Inception v3 model.

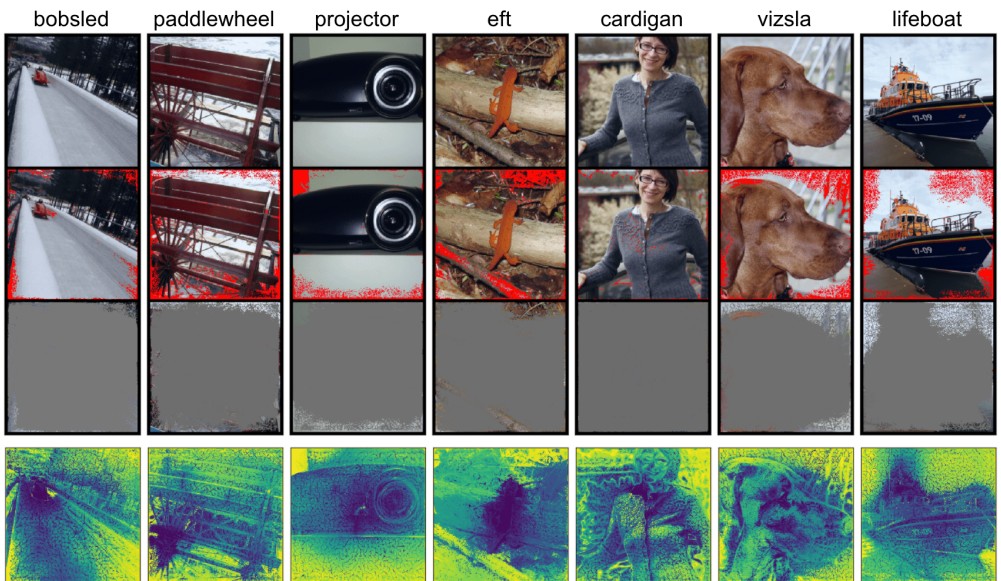

Figure S14: SIS subsets and ordering of pixels removed by Batched Gradient FindSIS in a sample of ImageNet validation images that are confidently ($\geq 90\%$) and correctly classified by the Inception v3 model. The top row shows original images, second row shows the location of SIS pixels (red), and third row shows images with all non-SIS pixels masked (and are still classified correctly with $\geq 90\%$ confidence). The heatmaps in the bottom row depict the ordering of batches of pixels removed during backward selection (blue = earliest, yellow = latest).

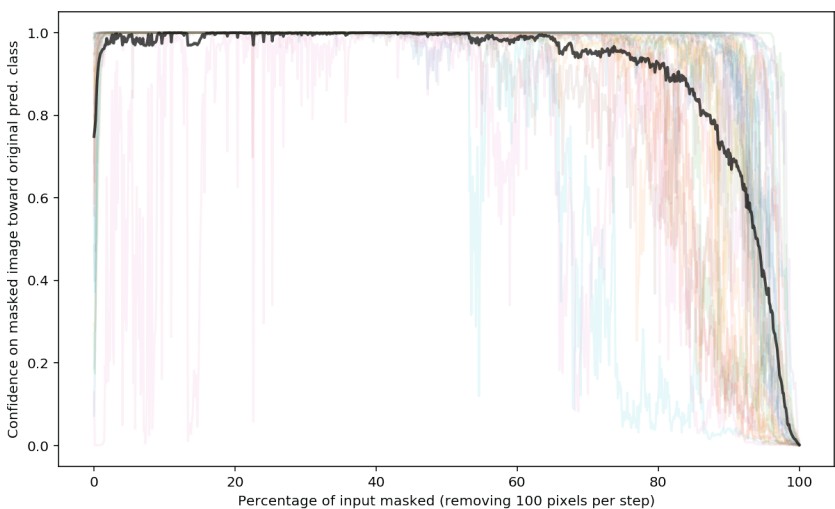

Figure S15: Prediction history on remaining (unmasked) pixels at each step of the Batched Gradient BackSelect procedure for a random sample of 32 ImageNet validation images by the Inception v3 model. Black line depicts mean confidence at each step.

### S6.3 SIS Size by Class

Figure S16 shows the distribution of SIS sizes by predicted class (SIS threshold 0.9) for all ImageNet validation images classified with $\geq$ 90% confidence (23080 images) by the pre-trained Inception v3.

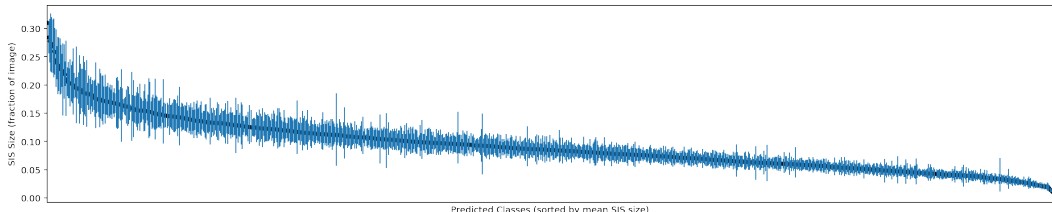

Figure S16: Mean SIS size per predicted ImageNet class by a pre-trained Inception v3 on ImageNet computed on ImageNet validation images (SIS threshold 0.9). Classes are sorted by mean SIS size. 95% confidence intervals are indicated around each mean. The top 5 classes with largest mean SIS size (mean % of image) are: English foxhound (40.0%), bee eater (28.4%), trolleybus (27.7%), Japanese spaniel (27.3%), whippet (27.0%). The 5 classes with the smallest mean SIS size are: bearskin (1.1%), bath towel (1.3%), wallet (1.4%), fire screen (1.7%), coffeepot (1.9%).

### S6.4 SIS for Vision Transformers

We applied Batched Gradient SIS to a vision transformer (ViT) [51] as ViTs have been shown to be more robust to perturbations and shifts than CNNs [52]. We used a pre-trained `B_16_imagenet1k` ViT model available from GitHub[7], which we found achieves 83.9% top-1 ImageNet validation accuracy. Figure S17 shows an example of the resulting SIS, suggesting this ViT likewise suffers from overinterpretation on ImageNet data.

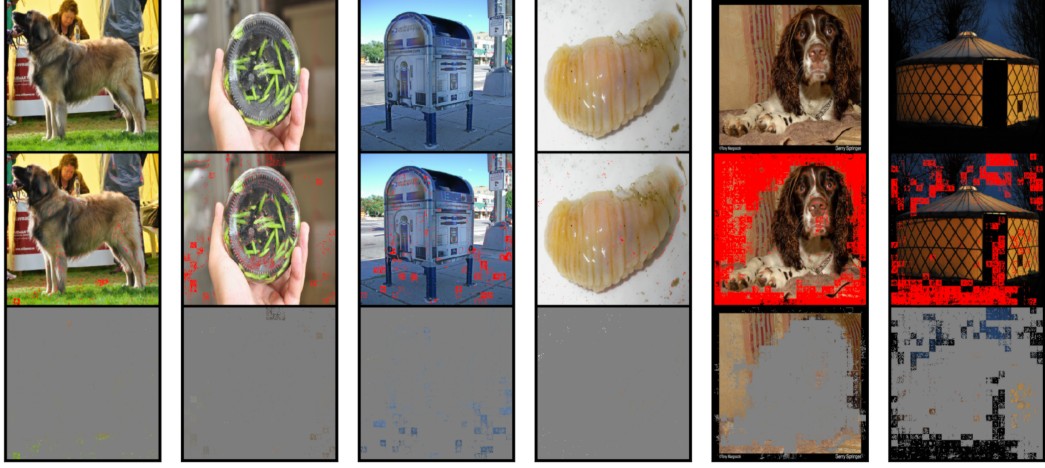

Figure S17: Example SIS (threshold 0.9) from ImageNet validation images (top row of each block) for a vision transformer (ViT). The middle rows show the location of SIS pixels (red) and the bottom rows show images with all non-SIS pixels masked but are still classified by the ViT model with $\geq$ 90% confidence.

---

[7]https://github.com/lukemelas/PyTorch-Pretrained-ViT