# OpenReview forum: "Overinterpretation reveals image classification model pathologies"
_NeurIPS.cc/2021/Conference — NeurIPS 2021 Poster_

### Official Review · Reviewer_LoQm · 2021-07-07

**Rating:** 8
**Confidence:** 4

**Summary:**

The paper identifies sparse sets of pixels which can be used with deep neural nets to classify images with high confidence. The authors demonstrate results with CIFAR-10, an augmentation corrupted version of it and with ImageNet. They show a number of experiments to obtain insights in why these spurious subsets are classified with high confidence.

For example, they show that the sparse subsets can be used to train a different model to predict well on CIFAR-10 and on Imagenet.
Furthermore they show that the SIS sizes are larger on correctly classified images compared to misclassified images, which implies that the sets are indeed used to properly predict on images.

**Ethics Review Area:**

["I don’t know"]

**Limitations And Societal Impact:**


Societal Impact:

This paper shows that CIFAR-10 is similar to MNIST a dataset suffering from simplicity, and results on it should be taken with care when it comes to transferability to other more complex datasets. It can be assumed that not every researcher will like to hear this kind of news. So it is worth to bring this in NeurIPS.

Limitations:

One should give a justification why batchgradSIS is chosen over e.g. optimizing masks and occlusion methods (or integrated gradients?) . Reviewer guesses likely due to fast runtime, or maybe the authors were interested in a greedy optimization which does not have strong connectedness priors on the remaining unmasked regions for the sake of mining unconnected sets?

Likely the noisyness of gradients (Balduzzi, The Shattered Gradients Problem: If resnets are the answer, then what is the question? , 2017) probably also helps in finding disconnected subsets.


The work can be put in a wider context of adversarial optimization, although it does not aim at small changes in the input but rather semantically meaningless examples as the original "neural networks are easily fooled" paper by AM Nguyen.

see also the missing citations in the box before.

**Main Review:**

clarity: it is an easy read, well written.

originality: The novelty of batchgradientSIS is limited. It is a natural iterative extension of the brute force SIS optimization. The paper would be more straightforward if one stressed the understanding aspect rather than the novelty of batchgradientSIS.

The existence of sparse subsets has not been analyzed so far, though shape-based background-artefacts in Imagenet have been observed in Finding and Removing Clever Hans: Using Explanation Methods to Debug and Improve Deep Models  (see some later figures in this paper).
The paper has a novelty in discussing pathologies of neural nets.

quality:
The experiments are well designed, to analyze various hypotheses, e.g. using sparse subsets discovered by one model to train another model and obtain similar accuracy.

as criticism points on quality:

missing citations:

-Regarding optimizing masks to retain minimal examples: Fong, Patrick, Vedaldi, Understanding Deep Networks via Extremal Perturbations and Smooth Masks ICCV2019
-By computing attributions and using attribution-score to guide iterative masking it belongs to a class of approaches which were used for CNNs in Samek et al, Evaluating the visualization of what a deep neural network has learned, TNNLS 2017
-Occlusion methods - e.g. Agarwal, Nguyen ACCV 2020
-Pertinent positives, Dhurandhar et al. 2018. While this method may have some degrees of freedom and ambiguities, it can be still mentioned.

significance: this paper adds a new angle to the discussion on deep neural network pathologies.

Despite limited novelty in methods, the reviewer finds the paper acceptable because it discusses a statistical defect of NNs from a new angle, and thus increases the understanding of existing algorithms rather than proposing a novel algorithm.


**Time Spent Reviewing:**

2.5 hours

---

> ### Author Response · Authors · 2021-08-10
> **Response to Reviewer LoQm**
>
> We thank the reviewer for the feedback and interest in our work.
>
> **[Citations]**  We thank the reviewer for pointing out these citations and will add all references to Section 2 of the revised paper.
>
> **[Choice of Batched Gradient SIS]**  Revealing overinterpretation requires a systematic way to identify the features used by a model to reach its decisions.  As the reviewer points out, it is important that the extracted rationales not be biased toward human visual priors, and that the method faithfully report the features used by a model (as discussed in lines 39-48 and 64-66).  We introduced Batched Gradient SIS to scale the backward selection procedure of SIS to ImageNet and enable discovery of overinterpretation on ImageNet.  All rationales discovered by SIS and Batched Gradient SIS are guaranteed to be sufficient for classification by a model.
>
> **[Wider context of adversarial optimization]**  We agree that our method could be viewed as an alternative way of generating a new type of adversarial example which does not explicitly perturb the image in an adaptive direction.  Existing adversarial methods synthesize new images or add extra information to images, whereas we demonstrate overinterpretation with unmodified subsets of images.  We found that adversarially robust models also suffer from overinterpretation (examples shown in Figure 1).  We also show training new classifiers on 5% of pixels produces surprisingly high test accuracy, which has not been shown for adversarial examples.  This result reveals degenerate signals in datasets, often unrelated to salient features, and thus models trained on these datasets may generalize poorly to real-world data.  Unlike adversarial methods, an application of our methods is evaluation of training sets to ensure decisions are made on interpretable signals as overinterpretation may be difficult to overcome with ML methods alone.

---

> > ### Comment · Reviewer_LoQm · 2021-09-01
> > **Thank you for the response**
> >
> > The reviewer feels ok with it.

---

### Official Review · Reviewer_dNig · 2021-07-14

**Rating:** 7
**Confidence:** 5

**Summary:**

The paper presents an interesting finding that CNNs overinterpret a few non-meaningful pixels in the image to make accurate predictions. The authors use SIS and extend it to Batched Gradient SIS to discover the limited number of pixels such CNNs rely on. The authors conduct extensive experiments on two major image datasets to verify the finding. Besides, ensembling and input dropout are explored to mitigate the overinterpretation problem.

**Limitations And Societal Impact:**

Although the ensembling and input dropout methods do not completely mitigate the overinterpretation issue (L471-474), the paper’s major focus is revealing and analyzing the overinterpretation problem. It is acceptable to leave it for future work. Therefore, the paper’s limitation is well-addressed. In terms of the potential negative societal impact, I do not think “training datasets may become more complex” is negative. In fact, removing such spurious signals in the dataset can make the model more robust, leading to a positive impact. Therefore, I do not think the paper has potential negative impacts because the paper tries to reduce them.

**Main Review:**

**Originality**: One of my concerns is the novelty of the method part. The proposed Batched Gradient SIS method seems to be a straightforward extension to SIS for better efficiency.

**Quality**: The paper contains extensive experiments on two image datasets to verify the overinterpretation problem (e.g., detailed analysis in Sec. 4.1 - 4.4 provides). One minor concern is that SIS pixels’ patterns are different between Fig. 1 and Fig. 3. Is it due to different datasets or due to different SIS algorithms (vanilla SIS vs. Batched Gradient SIS). Adding an experiment to show the SIS pixels from Batched Gradient SIS method on the CIFAR-10 dataset may explain that difference.

**Clarity**: The paper is well-written and easy to follow.

**Significance**: The finding of the paper, i.e., overinterpretation, is significant, which can facilitate future research to solve the overinterpretation problem for achieving more accurate and robust (not relying on spurious pixels) image classification models.

In summary, although the paper’s method is not novel, I believe that the paper’s main focus, i.e., revealing and analyzing the overinterpretation problem, does make a good contribution to the research community. Thus my rating is “Marginally above the acceptance threshold” to this paper.

**Time Spent Reviewing:**

3

---

> ### Author Response · Authors · 2021-08-10
> **Response to Reviewer dNig**
>
> We thank the reviewer for the feedback and interest in our work.
>
> **[Differences in SIS patterns in Figures 1 and 3]**   Further to the suggestion of the reviewer, we applied Batched Gradient SIS to CIFAR-10 images for all three architectures.  Examples of SIS subsets for ResNet18 from Batched Gradient SIS (confidence threshold 0.99, $k$ = 1) on CIFAR-10 test images, and a heatmap showing SIS pixel locations over all CIFAR-10 test images, are shown here: <https://i.imgur.com/2KT9KnK.png>.  We found Batched Gradient SIS produced edge-heavy heatmaps for CIFAR-10.  Heatmap differences are a result of the different valid equivalent SIS subsets found by the two SIS discovery algorithms.  However, since all SIS subsets are validated with a model and guaranteed to be sufficient for classification at the specified threshold, the heatmaps are accurate depictions of what is sufficient for the model to classify images at the threshold.  Overinterpretation is independent of the SIS algorithm used because both algorithms produce human-uninterpretable sufficient subsets (as shown in the examples).  We will include these results in the revised paper.
>
> **[Novelty]**  In addition to the introduction of Batched Gradient SIS, another contribution of our paper is the discovery and analysis of overinterpretation, and observation that training and testing on 5% of images is sufficient to provide comparable performance to full images (full list of contributions provided in L70-81).  Our paper presents a scientific investigation of this important phenomenon.  We find overinterpretation is a common failure mode of ML models, which latch onto non-salient but statistically valid signals in datasets, and we appreciate the reviewer’s finding the discovery of overinterpretation to be a good contribution to the community.

---

> > ### Comment · Reviewer_dNig · 2021-08-24
> > **Concerns are addressed**
> >
> > I appreciate the authors’ response. I have read all reviews and read the paper again. I think my concerns on 1) the difference of SIS pixel patterns between Fig. 1 and Fig. 3; and 2) novelty. I encourage the authors to include additional visualization in the final version.
> >
> > Since my concerns are resolved, I will increase my rating to 7.

---

### Official Review · Reviewer_eccV · 2021-07-15

**Rating:** 6
**Confidence:** 4

**Summary:**

Authors define "overinterpretation" as an undesired behavior in deep neural networks where the networks find strong class evidence in semantically unrelated image regions. The behaviour is quantified by identifying the minimal number of pixels sufficient to predict a given class in an input image while masking all the other features/pixels. In this regard, protocols are proposed to evaluate overinterpretation and the models trained on CIFAR and ImageNet are shown to suffer from this problem.

**Ethical Concerns:**

None.

**Limitations And Societal Impact:**

The limitations and societal impact is appropriately discussed in the paper.

**Main Review:**

Pros:
+ A batched gradient approach is developed to discover sufficient subsets in the input images.
+ Semantically irrelevant features contribute towards the prediction of true class in an image, this behavior is analyzed systematically. An interesting result is that masking ratios of up to 90-95% still lead to comparable test accuracies. To the best of my knowledge, this has not been shown earlier in this context.
+ Approaches to mitigate overinterpretation are also analyzed with some useful recommendations such as model ensembling.
+ Good number of experiments are performed to study the overinterpretation problem. The analysis also shows that the SIS size is related to the model performance on clean images.
+ Paper is clearly written with nice illustrations and visualizations.

Cons/Questions:
- The term SIS (sufficient input subsets) [4] is not defined before its use in Line 11.
- ResNet and VGG family models are explored to study overinterpretation. Some recent works (e.g., [A]) in the literature study randomly masking inputs and found ViTs to have much better robustness compared to CNNs.  It will be interesting to see how the robustness holds against masks learned in a white-box setting, similar to the setting studied in this work.
- It is important to understand here if the SIS patterns are specific to the input data distribution/image or the target model. There is one experiment on this point in the supplementary material for CIFAR10 dataset where different architectures are tried out with the same SIS. Due to the significance of this expeirment, the authors can improve it further with an analysis on a large dataset like ImageNet or across an altogether different family e.g., ViTs (as an extension to the above comment).
- The human study is performed with 4 participants. Were they shown sufficient examples of 5% SIS images to learn about the 10 CIFAR classes. Without such a precursor learning task, this experiment seems to be biased towards the model since SIS are found using the model in a white-box setting and therefore would make sense to the model but not for humans. A revserse eperiment has also been done in the literature [13] where humans can easily classifly textureless objects while DNN models suffer.

[A] Naseer et al., "Intriguing Properties of Vision Transformers", Arxiv 2021.

**Time Spent Reviewing:**

6

---

> ### Author Response · Authors · 2021-08-10
> **Response to Reviewer eccV**
>
> We thank the reviewer for the feedback and interest in our work.  We will improve the writing for clarity where indicated by the reviewer.
>
> **[Vision transformers]**  In response to this suggestion from the reviewer, we applied Batched Gradient SIS to ImageNet images using a pre-trained ViT (B_16_imagenet1k, 83.9% top-1 ImageNet validation accuracy).  We found that the resulting SIS subsets were unsalient, as shown in the following examples (of a random sample of ImageNet validation images that are classified with >= confidence 0.9 by the ViT): <https://i.imgur.com/VJe2hPG.png>.  All resulting SIS subsets shown (bottom row) are also classified by the pre-trained ViT with confidence >= 0.9.  We will include these results in our revised paper.
>
> **[Are SIS patterns specific to data distribution or model?]**  We repeated the experiment of Table S2 and found a similar result on ImageNet data.  We trained a DenseNet-121 model on 10% BS pixel-subsets of ImageNet training images drawn from the ResNet50, and the DenseNet-121 achieved 65.74% accuracy on the corresponding 10% BS pixel-subsets of ImageNet validation images from ResNet50, consistent with ResNet50 accuracies reported in Table S4.   We will add these results to our paper.
>
> **[Human study]**  Humans were not shown examples of 5% pixel-subsets prior to the study.  The goal of our human study is to demonstrate that humans cannot discern salient features in sparse pixel-subsets relied upon by models for confident classification of images.  We assume humans are the gold standard for annotation of images and thus for determining the salient features of an image.  Our results thus show that vision models can rely on features that humans consider non-salient, which can lead to poor generalization.

---

> > ### Comment · Reviewer_eccV · 2021-08-19
> > **Final Comments**
> >
> > I thank the authors for providing satisfactory answers to my queries and interesting new results. I would suggest to include these results to the revised paper which will hopefully help further improve the paper quality.

---

### Official Review · Reviewer_cYV5 · 2021-07-16

**Rating:** 7
**Confidence:** 4

**Summary:**

The paper evaluates pathologies of modern neural networks on CIFAR-10 and ImageNet. Specifically, the authors show that these networks exhibit *overinterpretation*, which is a phenomenon where a model can use sparse subsets of an image which are semantically meaningless to humans in order to confidently and accurately classify the image. The authors utilize the Sufficient Input Subsets (SIS) algorithm from the interpretability literature to identify these subsets, but they also propose modifications to this algorithm which allow it to scale to ImageNet. Surprisingly, the authors find that models evaluated on a small subset of SIS pixels from each image (5% on CIFAR-10 and 10% on ImageNet) exhibit only a small drop in accuracy. The authors explore two techniques for mitigating overinterpretation: ensembling and input dropout.

**Limitations And Societal Impact:**

The authors discuss limitations briefly, but do not address negative societal impacts.

**Main Review:**

**Originality** Though the SIS algorithm is not new, the authors use it to make important and novel observations about neural network behaviors. They also modify the algorithm to be applicable to ImageNet.

**Quality**:  Overall, the submission is technically sound and the experimental results are convincing. However, I have some comments I hope the authors can address:

**Major comments**

1. *Calibration of models*. Since neural networks are known to be poorly calibrated and over confident, thresholding the model confidences based on a fixed parameter may lead to inconsistent results across uncalibrated models. Are the models calibrated before applying SIS?
2. *Use of confidence instead of accuracy in SIS algorithm*.  Would it be possible to set a threshold on the model accuracy instead of the model confidence in the SIS algorithm?  The size of the subset needed to reach a certain accuracy threshold on both in-distribution and out-of-distribution datasets might be a more meaningful way to compare the performance of different models.
3. *Evaluating on full images when training on subsets* In Table 1, it would be interesting to compare the model accuracy on the full images when trained on either the BS or random subsets in order to understand how much training on the subsets hurts model performance.
4. *Size of the subset versus model accuracy* . The authors claim that accuracy and SIS size are correlated for ensembles, but not for models trained with input dropout. For regular models, it’s not clear what the trend is. It would be interesting to see model accuracy on both in-distribution and out-of-distribution data versus SIS size.
5. *Evaluation on CIFAR-10.1*  CIFAR-10C is a synthetic distribution shift whereas CIFAR-10.1 (Recht et. al. 2018) is a natural distribution shift. Natural distribution shifts are more realistic than synthetic shifts. Moreover, techniques that work well on synthetic shifts do not necessarily transfer to natural distribution shifts (Taori et. al. 2020). The authors should evaluate out-of-distribution robustness on CIFAR-10.1 in addition to CIFAR-10C.

**Minor comments**

6. *Explanation for heatmaps* Can the authors comment on why the ResNet20 heat map is so different from the ResNet18 and VGG16 heat map? Also, why are the ImageNet subsets concentrated on the border of the images, but the same is not true for CIFAR?
7. *SIS size per class* Does the size of the SIS subset vary across ImageNet classes?
8. *Conflicting observations* The observations on lines 267-273 seem to directly conflict with the observations on lines 226-233. Can the authors clarify?

**Significance** The paper shows a way to detect spurious features that a model over-relies on when making predictions.  This has significant implications for robustness research which aims to prevent such over-reliance.


**Time Spent Reviewing:**

3

---

> ### Author Response · Authors · 2021-08-10
> **Response to Reviewer cYV5**
>
> We thank the reviewer for the feedback and interest in our work.
>
> **[1. Calibration of models]**
> We discussed potential issues of model calibration in our paper (L256 and L298).   The models were trained using standard techniques and were not calibrated before applying SIS.  In response to the reviewer, we ran additional experiments and calibrated our CIFAR-10 models.  We calibrated each model after training using Temperature Scaling (Guo et al., 2017) and found that while the SIS size (for test images that the re-calibrated model can classify with >= 0.99 confidence) does increase slightly, the resulting SIS subsets are still semantically meaningless and far below the threshold of SIS size where humans can meaningfully start to classify CIFAR images with any degree of accuracy.  For example, for ResNet20, temperature scaling improved expected calibration error (ECE) on test images from 3.91% (uncalibrated) to 0.91% (calibrated).  Mean SIS size increased from 2.36% of images to 2.94% of images, and the number of test images classified with >= 0.99 confidence decreased from 7829 to 5805 after calibration.  SIS subsets still appear semantically meaningless, and we have included a sample of SIS subsets classified with >= 0.99 confidence by the calibrated model here: <https://i.imgur.com/wjz6SwZ.png>.  We will include full results in the supplement of our final paper.  We also point out that one of the key findings of our paper is that even when we compute SIS subsets from uncalibrated models, those subsets still contain enough signal for training entirely new classifiers that can generalize as well to the corresponding test subsets.
>
> **[2. Use of confidence instead of accuracy in SIS algorithm]**
> We first clarify that for an already-trained model and any valid SIS subset of any image with respect to this model, the model’s predictions of the class-label (rather than class-probability) always remain the same for the SIS subset vs the full image (by the definition of valid SIS). Thus the test accuracy of the already-trained model will remain exactly the same if it is only fed the semantically meaningless SIS-subsets rather than full images in the test set.
>
> Suppose we redefine the SIS-threshold $\tau$ adaptively per image to equal $\max_{\text{class}} P(\text{class} | \text{image})$, i.e., the smallest $\tau$ such that the model’s accuracy is exactly preserved (because the model would still predict the same class-label for the SIS-subset at this $\tau$ as for the original image). Under this alternative, the values of $\tau$ are far lower for most individual images that than values used in our paper, and therefore the SIS subsets are even smaller with even less semantic information than the ones shown in our paper (i.e., the degree of overinterpretation becomes more severe). Thus this alternative accuracy-threshold does not qualitatively change our conclusions and only highlights the issue of overinterpretation to an even greater degree.
>
> Furthermore, our primary goal in this paper is to share two major insights -- (1) overinterpretation is not just a ‘model’ issue but an issue with the benchmarks where they contain meaningless pixel subsets that are still statistically sufficient to train high accuracy models, and (2) inductive biases of existing models and regularization/training techniques (including models trained adversarially against standard perturbations) are insufficient to mitigate the overinterpretation issue. We believe that the existing methodology/experiments used in the current paper clearly illustrate these two insights with substantial evidence to support the claims.
>
> **[3. Evaluating on full images when training on subsets in Table 1]**
> We included these results in Supplementary Table S5 (L607-613).  While accuracies are generally significantly higher than random guessing, we note that full images are highly out-of-distribution for a model trained on images with only 5% pixel-subsets and hence such a model cannot properly generalize to full images.  Further, the model trained on 5% images may not rely on the same features as the model trained on full images as it is trained on a substantially different training set.
>
> **[4. Size of the subset versus model accuracy]**
> Mean SIS size for all models (including ResNet18, ResNet20, and VGG16) at varying SIS thresholds are shown in Figure 5.  The accuracies of these models for CIFAR-10 test data are shown in the Figure 5 legend, and in Table 1 for CIFAR-10-C OOD data.
>
> **[5. Evaluation of CIFAR-10.1]**
> We have additionally evaluated our CIFAR-10 models on CIFAR-10.1 v6 images, and the results are provided at: <https://i.imgur.com/VFZxZR4.png>.  We will include these results on CIFAR-10.1 in the revised paper.
>
> **[6. Explanation for heatmaps]**
> We believe that the concentration of SIS pixels on the bottom border for ResNet20 (Figure 2a) is a result of tie-breaking during backward selection of the SIS procedure.  To explore this hypothesis, we modified the tie-breaking procedure to randomly (rather than deterministically) break ties during SIS backward selection and found the resulting SIS heatmap more concentrated in the image centers rather than bottom border.  We also ran Batched Gradient SIS on CIFAR-10 images for ResNet18 and found Batched Gradient SIS produced edge-heavy heatmaps for CIFAR-10 (see examples and heatmap linked in response below).  Both of these results will be included in the revised paper.  These heatmap differences are a result of the different valid equivalent SIS subsets found by the two SIS discovery algorithms.  However, since all SIS subsets are validated with a model and guaranteed to be sufficient for classification at the specified threshold, the heatmaps are accurate depictions of what is sufficient for the model to classify images at the threshold.  Overinterpretation is independent of the SIS algorithm used because both algorithms produce human-uninterpretable sufficient subsets (as shown in the examples).
>
>
> **[7. SIS size per class]**
> Further to the suggestion of the reviewer, we computed the distribution of SIS size per class for CIFAR-10 (SIS threshold 0.99) and ImageNet (SIS threshold 0.9) and found that the SIS subset size does differ per class for both datasets.  We have uploaded these figures anonymously at: <https://i.imgur.com/Hw8d7IY.png> (CIFAR-10), <https://i.imgur.com/1CElK5h.png> (ImageNet, Inception-v3).  For ImageNet, the top 5 classes with largest mean SIS size are: English foxhound, bee eater, trolleybus, Japanese spaniel, whippet.  The 5 classes with the smallest mean SIS size are: bearskin, bath towel, wallet, fire screen, coffeepot.   We will add these results to the supplement of our revised paper.
>
> **[8. Conflicting observations]**
> Our observation on L226-233 is that SIS subsets confidently classified by one model (trained on full CIFAR-10 images) are not accurately classified by other models (of the same or different architecture class) that were independently trained on full CIFAR-10 images (with different random initialization).  This finding suggests that there exist multiple statistical patterns that models trained on CIFAR-10 may learn to rely on, and that CIFAR-10 image classification is an underdetermined problem.   Our results on L267-273 show that we can train *new* models of any architecture class on 5% pixel-subsets of CIFAR-10 images that generalize well to the corresponding 5% pixel-subsets of test images, regardless of the model class employed for subset discovery.  The full results of this experiment are included in Table S2, and we have further found this hypothesis supported for ImageNet models as discussed below.

---

> > ### Comment · Reviewer_cYV5 · 2021-08-26
> > **Response**
> >
> > Thank you for answering my questions and running so many additional experiments. I appreciate the diligence that went into the response and feel the additional results significantly strengthen the paper. I am raising my score to a 7.

---

### Public Comment · ~Liang_Liang2 · 2022-01-19
**high confidence for OOD samples**

we have a different explanation about high confidence for OOD samples
see
https://arxiv.org/abs/2009.08016
https://github.com/liangbright/OOD_Attack_NN

---

### Decision · Program_Chairs · 2021-09-27

**Decision:**

Accept (Poster)

**Comment:**

This work argues that deep neural networks overinterpret the input, i.e. they rely on too small portions of the image to make the decision. This is novel insight and I am enthusiastic that it will help the community understand better deep networks. All reviewers agree that the paper should be accepted, and I am glad to support this decision. Please remember to address all reviewers' comments in the camera-ready version.